# Medformer: A Multi-Granularity Patching Transformer for Medical Time-Series Classification

**Yihe Wang**[*]**, Nan Huang**[*]**, Taida Li**[*]
University of North Carolina - Charlotte
{ywang145,nhuang1,tli14}@charlotte.edu

**Yujun Yan**
Dartmouth College
yujun.yan@dartmouth.edu

**Xiang Zhang**
University of North Carolina - Charlotte
xiang.zhang@charlotte.edu

## Abstract

Medical time series (MedTS) data, such as Electroencephalography (EEG) and Electrocardiography (ECG), play a crucial role in healthcare, such as diagnosing brain and heart diseases. Existing methods for MedTS classification primarily rely on handcrafted biomarkers extraction and CNN-based models, with limited exploration of transformer-based models. In this paper, we introduce Medformer, a multi-granularity patching transformer tailored specifically for MedTS classification. Our method incorporates three novel mechanisms to leverage the unique characteristics of MedTS: cross-channel patching to leverage inter-channel correlations, multi-granularity embedding for capturing features at different scales, and two-stage (intra- and inter-granularity) multi-granularity self-attention for learning features and correlations within and among granularities. We conduct extensive experiments on five public datasets under both subject-dependent and challenging subject-independent setups. Results demonstrate Medformer's superiority over 10 baselines, achieving top averaged ranking across five datasets on all six evaluation metrics. These findings underscore the significant impact of our method on healthcare applications, such as diagnosing Myocardial Infarction, Alzheimer's, and Parkinson's disease. We release the source code at https://github.com/DL4mHealth/Medformer.

## 1 Introduction

Medical time series refers to sequences of health-related data points recorded at successive times, tracking various physiological signals over time [1, 2]. Effective classification of MedTS data enables continuous monitoring and real-time analysis of a subject's physiological state, supporting early abnormality detection, accurate diagnosis, timely intervention, and personalized treatment—ultimately enhancing patient outcomes and healthcare efficiency [3, 4]. For instance, Electroencephalography (EEG) provides insights into a subject's neurological status [5, 6], while Electrocardiography (ECG) aids in diagnosing heart conditions [7, 8, 9]. Most current MedTS classification approaches rely on handcrafted biomarker extraction [10, 11, 12], convolutional neural networks (CNN)-based models [13, 14, 15, 16], graph convolutional networks (GNNs)-based models[17, 18], or combinations of CNNs and self-attention modules[19, 20]. Notably, effective transformer-based models for MedTS classification remain underexplored.

Transformers have demonstrated strong performance in time series representation learning across tasks such as forecasting [21, 22, 23], classification [24, 25], and anomaly detection [26, 27], with

---

[*]These authors contributed equally to this work.

38th Conference on Neural Information Processing Systems (NeurIPS 2024).

a predominant focus on forecasting. While these methods are applicable to MedTS classification, their design motivations and mechanisms may not fully align with the unique requirements of this domain. For example, as shown in Figure 1, models like Autoformer [28] and Informer [29] adopt the token embedding approach from the vanilla transformer [30], using a single cross-channel timestamp as an input token. This strategy struggles to capture coarse-grained temporal features. In contrast, iTransformer [31] encodes the entire series from one channel as an input token, which can overlook fine-grained temporal features while focusing on multi-channel correlations. Additionally, PatchTST [32] embeds a sequence of timestamps from one channel as a patch for self-attention, limiting the model's capacity to learn cross-channel relationships.

These existing methods fail to fully exploit the distinctive characteristics of MedTS data, such as local temporal dynamics, inter-channel correlations, and multi-scale feature analysis. First, effective capturing temporal dynamics demands multi-timestamp inputs to capture local temporal patterns, as highlighted in approaches like PatchTST [32] and Crossformer [33]. Second, leveraging cross-channel information is critical; for example, multi-channel EEG data recorded following the International 10–20 system [34] monitors the brain activities, with each electrode/channel corresponding to specific brain regions. Since brain functions are integrated, inter-channel correlations (e.g., brain connectome) are crucial in EEG analysis [35, 36, 37]. Third, representation learning across multiple temporal scales and periods is vital to uncover a broad range of health patterns, as certain disease indicators may only appear within specific frequency bands [10, 12].

To bridge this gap, we propose **Medformer**[*], a multi-granularity patching transformer designed explicitly for MedTS classification. Our approach introduces three mechanisms to enhance learning capacity. First, we propose a novel token embedding method using cross-channel patching, effectively capturing both *multi-timestamp* and *cross-channel* features. To the best of our knowledge, this is the first application of cross-channel patching for transformer embedding in time series analysis. Second, rather than using fixed-length patches, we employ *multi-granularity* patching with a list of patch lengths, enabling the model to capture features in different scales. This multi-granularity approach could simulate different frequency bands, capturing band-specific features without relying on hand-crafted up/downsampling and band filters. Third, We introduce a two-stage (intra- and inter-granularity) self-attention mechanism to capture features within individual granularities and correlations across granularities, enabling complementary information integration across scales.

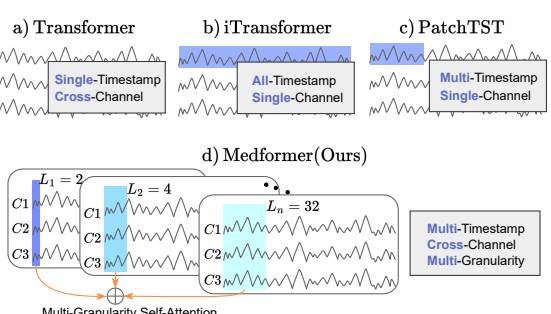

Figure 1: **Token embedding methods.** Vanilla transformer, Autoformer, and Informer [30, 28, 29] employ a single cross-channel timestamp as a token; iTransformer [31] utilizes an entire channel as a token; and PatchTST and Crossformer [32, 33] adopt a patch of timestamps from one channel as a token. For MedTS classification, we propose Medformer considering inter-channel dependencies (cross-channel), temporal properties (multi-timestamp), and multifaceted scale of temporal patterns (multi-granularity).

We conduct extensive experiments using ten baselines across five public datasets, including three EEG datasets and two ECG datasets, focused on detecting diseases such as Alzheimer's and cardiovascular conditions under both subject-dependent and subject-independent setups (Figure 2). Results show that Medformer achieves the highest average ranking across all six evaluation metrics and five datasets (Figure 4), highlighting its superior effectiveness, stability, and potential for real-world applications.

## 2 Related Work

**Medical Time Series.** Medical time series refers to specialized time series data collected from the human body, commonly used for disease diagnosis [3, 7], health monitoring [6, 1], and brain-computer interfaces (BCIs) [39, 2]. Different MedTS modalities include EEG [40, 41, 42], ECG [7, 8, 9], EMG [43, 44], and EOG [45, 46], each offering distinct capabilities for various medical applications.

---

[*]We note that this name has been previously used in other domains [38].

For example, EEG and ECG data are instrumental in assessing brain and heart health [40, 7]. Recent BCI research explores using EEG to control objects, providing functional support to individuals with disabilities [2, 47]. Unlike general time series research, which predominantly focuses on forecasting tasks [48, 49], MedTS research is centered around signal decoding, which involves classifying hidden information within MedTS sequences. Current approaches often rely on biomarker identification and deep-learning models utilizing CNNs, GNNs, or hybrid models combining CNNs with self-attention modules. For instance, band features such as relative band power and inter-band correlations [11, 50] have proven effective in EEG-based Alzheimer's disease diagnosis. Deep-learning models like EEGNet [14], EEG-Conformer [20], and TCN [51, 13] have also shown strong performance across various MedTS classification tasks.

**Transformers for Time Series.** Transformer has demonstrated its strong learning and scaling-up ability in many domains, including natural language processing [30, 57] and computer vision [58, 59]. Existing transformer-based methods for time series analysis can be categorized into two main directions: modifying token embedding methods and self-attention mechanisms, or both. For example, PatchTST [32] uses a sequence of single-channel timestamps as a patch for token embedding. Methods like Autoformer [28], Informer [29], Nonformer [54], and FED-former [52] develop new self-attention mech-

**Table 1: Existing methods do not fully utilize all potential characteristics in MedTS.**

| Models | Multi-Timestamp | Cross-Channel | Multi-Granularity | Granularity-Interaction |
|---|:---:|:---:|:---:|:---:|
| **Autoformer** [28] | | ✓ | | |
| **Crossformer** [33] | ✓ | ✓ | | |
| **FEDformer** [52] | | ✓ | | |
| **Informer** [29] | | ✓ | | |
| **iTransformer** [31] | ✓ | ✓ | | |
| **MTST** [53] | ✓ | | ✓ | |
| **Nonformer** [54] | | ✓ | | |
| **PatchTST** [32] | ✓ | | | |
| **Pathformer** [55] | ✓ | | ✓ | |
| **Reformer** [56] | | ✓ | | |
| **Transformer** [30] | | ✓ | | |
| **Medformer(Ours)** | ✓ | ✓ | ✓ | ✓ |

anisms or replace the self-attention module to improve learning ability and reduce complexity. Crossformer [33] and iTransformer [31] modify both token embedding methods and self-attention mechanisms. **Patching.** Patch embedding has been widely used in time series transformers since the proposal of PatchTST [32]. Existing methods of patching, such as Crossformer [33], CARD [23], and MTST [53], inherit from PatchTST [32] and utilize a sequence of single-channel timestamps for patching. This channel-independent patching might benefit learning ability in time series forecasting but may not be as effective in MedTS classification. **Multi-Granularity.** Existing methods such as Pyraformer [21], MTST [53], Pathformer [55], and Scaleformer [60], utilize multi-granularity embedding to capture features at different scales, allowing models to learn both fine-grained and coarse-grained patterns. We discuss the differences between our method and existing multi-granularity approaches in Appendix G.1.

Medformer includes both novel token embedding and self-attention mechanisms. Figure 1 and Table 1 present a comparison of token embedding methods and feature utilization between our method and existing methods. The components of our method can be easily incorporated into existing methods to improve classification learning ability. For example, cross-channel multi-granularity patching can be integrated with methods that modify self-attention mechanisms, such as Autoformer [28] and Informer [29], for token embedding. Similarly, the two-stage multi-granularity self-attention can be combined with existing multi-granularity methods, like MTST [53] and Pathformer [55], to enhance the learning of inter-granularity features.

## 3 Preliminaries and Problem Formulation

**Disease Diagnosis with MedTS.** Medical time series data typically exhibit multiple hierarchical levels, including subject, session, trial, and sample levels [13]. In disease diagnosis tasks using MedTS, each subject is usually assigned a single label, such as indicating the presence or absence of Alzheimer's disease. However, multi-labeling may be necessary when a subject has co-occurring conditions [61]. Notably, a subject's medical or physiological state remains relatively stable over time (or within short periods without significant change). For instance, a subject diagnosed with Alzheimer's disease is expected to retain that diagnosis for many years. If the subject also has Parkinson's disease, a multi-label learning approach is required, which essentially conducts classification tasks for each disease independently if they are not mutually exclusive.

Since long sequences of time series data (e.g., trials or sessions) are often segmented into shorter samples for deep learning training, all samples from a single subject should ideally retain the same

medical condition label. Thus, each MedTS sample typically includes a class label indicating a specific disease type and a subject ID indicating its original subject. Given the ultimate goal of diagnosing whether a subject has a particular disease, experimental setups must be carefully designed to reflect real-world clinical applications. Diverse setups can yield markedly different outcomes, potentially leading to misleading conclusions. Here, we introduce two widely used setups in MedTS classification and clarify their distinctions. Figure 2 provides a simple illustration of these two setups.

**Subject-Dependent.** In this setup, the division into training, validation, and test sets is based on time series **samples**. All samples from various subjects are randomly shuffled and then allocated into the respective sets. Consequently, samples with identical subject IDs may be present in the training, validation, and test sets. This scenario potentially introduces "information leakage," wherein the model could inadvertently learn the distribution specific to certain subjects during the training phase. This setup is typically employed to assess whether a dataset exhibits cross-subject features and has limited applications under real-world MedTS-based disease diagnosis scenarios. The reason is simple: we cannot know the label of unseen subjects and their corresponding samples during training. Generally, the results of the subject-dependent setup tend to be notably higher than those from the subject-independent setup, often showing the upper limit of a dataset's learning capability.

**Subject-Independent.** In this setup, the division into training, validation, and test sets is based on **subjects**. Each subject and their corresponding samples are exclusively distributed into one of the training, validation, or test sets. Consequently, samples with identical subject IDs can only be present in one of these sets. This setup holds significant importance in disease diagnosis tasks as it closely simulates real-world scenarios. It enables us to train a model on subjects with known labels and subsequently evaluate its performance on unseen subjects; in other words, evaluate if a subject has a specific disease. However, this setup poses significant challenges in MedTS classification tasks. Due to the variability in data distribution and the potential presence of unknown noise within each subject's data, capturing general task-related features across subjects becomes challenging [62, 13, 63, 64]. Even if subjects share the same label, the personal noise inherent in each subject's data may obscure these common features. Developing a method that effectively captures common features among subjects while disregarding individual noise remains an unsolved problem.

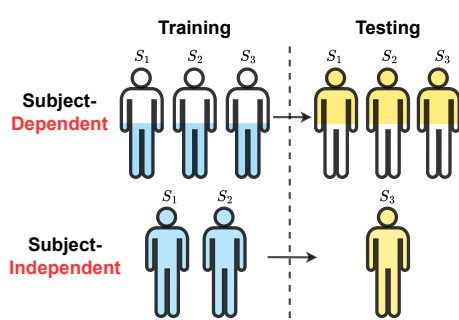

**Figure 2: Subject-dependent/independent setups (figure adopted from our previous work [13]).** In the subject-dependent setup, samples from the same subject can appear in both the training and test sets, causing information leakage. In a subject-independent setup, samples from the same subject are exclusively in either the training or test set, which is more challenging and practically meaningful but less studied.

In this work, we evaluate our model mainly in the subject-independent setup to better align with real-world applications, aiming to draw attention within the time series research community to the substantial challenges posed by this setup. *While our model is not specifically tailored to address subject-independent problems, it integrates multi-timestamp, cross-channel, and multi-granularity features in MedTS, enhancing its capacity to capture subject-invariant representations.* Consequently, our model is well-equipped to tackle the subject-independent challenge to a certain extent, and our results (Section 5) confirm such capability of Medformer.

We next present the problem formulation for multivariate medical time series classification in the context of disease diagnosis.

**Problem (MedTS Classification).** *Consider an input MedTS sample $\boldsymbol{x}_{in} \in \mathbb{R}^{T \times C}$, where $T$ denotes the number of timestamps and $C$ represents the number of channels. Our objective is to learn an encoder that can generate a representation $\boldsymbol{h}$, which can be used to predict the corresponding label $\boldsymbol{y} \in \mathbb{R}^K$ of the input sample. Here, $K$ denotes the number of medically relevant classes, such as various disease types or different stages of one disease.*

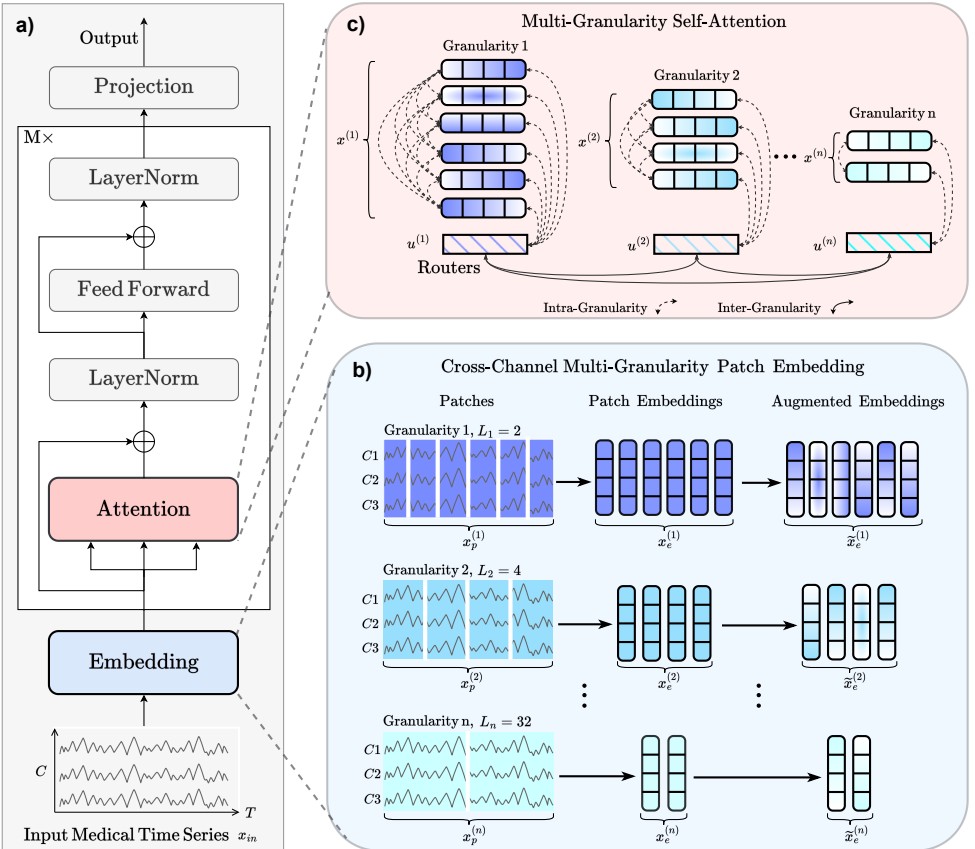

**Figure 3: Overview of Medformer.** a) Workflow. b) For the input sample $\boldsymbol{x}_{\text{in}}$, we apply $n$ distinct patch lengths in parallel to create patched features $\boldsymbol{x}_p^{(i)}$, where $i$ ranges from 1 to $n$. Each patch length represents a unique granularity. These patches are then projected into $\boldsymbol{x}_e^{(i)}$ and subsequently augmented to form $\widetilde{\boldsymbol{x}}_e^{(i)}$. c) We obtain the final embedding $\boldsymbol{x}^{(i)}$ by combining the augmented $\widetilde{\boldsymbol{x}}_e^{(i)}$ with both the positional embedding $\boldsymbol{W}_{\text{pos}}$ and the granularity embedding $\boldsymbol{W}_{\text{gr}}^{(i)}$. Additionally, a granularity-specific router $\boldsymbol{u}^{(i)}$ is designed to capture integrated information for each respective granularity. We then perform intra-granularity self-attention, focusing on individual granularities, and inter-granularity self-attention, using the routers to facilitate communication across different granularities.

## 4 Method

In this section, we first describe the cross-channel multi-granularity patching mechanism for learning spatial-temporal features in different scales. Next, we analyze the two-stage multi-granularity self-attention mechanism, which leverages features within the same granularity and correlations among different granularities. The architecture of the proposed Medformer is illustrated in Figure 3.

**Cross-Channel Multi-Granularity Patch Embedding.** From the medical perspective, the brain or heart functions as a cohesive unit, suggesting a naive assumption that there are inherent correlations among different channels in MedTS [35, 36, 37], as each channel represents the activities of distinct brain or heart regions. *Motivated by the above assumption*, we reasonably propose multi-channel patching for token embedding, which is different from existing patch embedding methods that embed patches in a channel-independent manner and fail to capture inter-channel correlations [32, 33, 23]. Figure 1 provides an overview comparison of existing token embedding methods and ours. Additionally, existing research on EEG biomarker extraction has shown that certain features are linked to different frequency bands, such as $\alpha, \beta$, and $\gamma$ bands [10, 12]. This *motivates* us to embed patch tokens in a multi-granularity way. Instead of using traditional methods like up/downsampling or handcrafted band filtering, multi-granularity patching automatically corresponds to different sampling frequencies, which can simulate different frequency bands and capture band-related features.

Given the above rationales, we propose a novel token embedding approach: cross-channel multi-granularity patching. Given an input multivariate MedTS sample $\boldsymbol{x}_{\text{in}} \in \mathbb{R}^{T \times C}$, and a list of different patch lengths $\{L_1, L_2, \ldots, L_n\}$. For the $i$-th patch length $L_i$ denoting granularity $i$, we segment the input sample into $N_i$ cross-channel non-overlapping patches $\boldsymbol{x}_{\text{p}}^{(i)} \in \mathbb{R}^{N_i \times (L_i \cdot C)}$. Zero padding is applied to ensure that the number of timestamps $T$ is divisible by $L_i$, making $N_i = \lceil T/L_i \rceil$.

The patches are mapped into latent embeddings space using a linear projection: $\boldsymbol{x}_{\text{e}}^{(i)} = \boldsymbol{x}_{\text{p}}^{(i)} \boldsymbol{W}^{(i)}$, where $\boldsymbol{x}_{\text{e}}^{(i)} \in \mathbb{R}^{N_i \times D}$ and $\boldsymbol{W}^{(i)} \in \mathbb{R}^{(L_i \cdot C) \times D}$. Inspired by the augmented views contrasting in the contrastive learning framework [65, 13, 66], we further apply data augmentations such as masking and jittering on $\boldsymbol{x}_{\text{e}}^{(i)}$ to obtain augmented embeddings $\widetilde{\boldsymbol{x}}_{\text{e}}^{(i)} \in \mathbb{R}^{N_i \times D}$. We assume the augmentation can improve the learning ability in the following inter-granularity self-attention stage by forcing different granularities to learn and complement information from each other.

A fixed positional embedding $\boldsymbol{W}_{\text{pos}} \in \mathbb{R}^{G \times D}$ is generated for positional encoding [30], where $G$ is a very large number. We add $\boldsymbol{W}_{\text{pos}}[1 : N_i] \in \mathbb{R}^{N_i \times D}$, the first $N_i$ rows of the positional embedding $\boldsymbol{W}_{\text{pos}}$, and a learnable granularity embedding $\boldsymbol{W}_{\text{gr}}^{(i)} \in \mathbb{R}^{1 \times D}$, to obtain the final patch embedding for the $i$-th granularity with patch length $L_i$:

$$\boldsymbol{x}^{(i)} = \widetilde{\boldsymbol{x}}_{\text{e}}^{(i)} + \boldsymbol{W}_{\text{pos}}[1 : N_i] + \boldsymbol{W}_{\text{gr}}^{(i)}, \tag{1}$$

where $\boldsymbol{x}^{(i)} \in \mathbb{R}^{N_i \times D}$. Note that the granularity embedding $\boldsymbol{W}_{\text{gr}}^{(i)}$ aims to distinguish among granularities and is broadcasted to all $N_i$ embedding rows with $D$ dimension during addition.

To reduce time and space complexity, we initialize a router to be used in the multi-granularity self-attention (as described later) for each granularity:

$$\boldsymbol{u}^{(i)} = \boldsymbol{W}_{\text{pos}}[N_i + 1] + \boldsymbol{W}_{\text{gr}}^{(i)}, \tag{2}$$

where $\boldsymbol{u}^{(i)}, \boldsymbol{W}_{\text{pos}}[N_i + 1], \boldsymbol{W}_{\text{gr}}^{(i)} \in \mathbb{R}^{1 \times D}$. Here, $\boldsymbol{W}_{\text{pos}}[N_i + 1]$ is not used for positional embedding but to inform the router about the number of patches with the current $L_i$ granularity, and $\boldsymbol{W}_{\text{gr}}^{(i)}$ contains the granularity information. Both components help distinguish the routers from one another.

Finally, we obtain a list of final patch embeddings $\{\boldsymbol{x}^{(1)}, \boldsymbol{x}^{(2)}, \ldots, \boldsymbol{x}^{(n)}\}$ and router embeddings $\{\boldsymbol{u}^{(1)}, \boldsymbol{u}^{(2)}, \ldots, \boldsymbol{u}^{(n)}\}$ for different granularities with patch lengths $\{L_1, L_2, \ldots, L_n\}$. We feed the embeddings to the two-stage multi-granularity self-attention.

**Multi-Granularity Self-Attention.** Our goal is to learn multi-granularity features and granularity interactions during self-attention. A naive approach to achieve this goal is to concatenate all the patch embeddings $\{\boldsymbol{x}^{(1)}, \boldsymbol{x}^{(2)}, \ldots, \boldsymbol{x}^{(n)}\}$ into a large patch embedding $\boldsymbol{X} \in \mathbb{R}^{(N_1 + N_2 + \ldots + N_n) \times D}$ and perform self-attention on this new embedding, where $n$ denotes the number of different granularities. However, this results in a time complexity of $O\left(\left(\sum_{i=1}^{n} N_i\right)^2\right)$, which is impractical for a large $n$.

To reduce the time complexity, we propose a router mechanism and split the self-attention module into two stages: a) intra-granularity self-attention and b) inter-granularity self-attention. The intra-granularity stage performs self-attention within the same granularity to capture the distinctive features of each granularity. The inter-granularity stage performs self-attention across different granularities to capture their correlations.

**a) Intra-Granularity Self-Attention.** For the $i$-th patch length $L_i$ denoting granularity $i$, we verti-cally concatenate the patch embedding $\boldsymbol{x}^{(i)} \in \mathbb{R}^{N_i \times D}$ and router embedding $\boldsymbol{u}^{(i)} \in \mathbb{R}^{1 \times D}$ to form an intermediate sequence of embeddings $\boldsymbol{z}^{(i)} \in \mathbb{R}^{(N_i + 1) \times D}$:

$$\boldsymbol{z}^{(i)} = \left[\boldsymbol{x}^{(i)} \| \boldsymbol{u}^{(i)}\right] \tag{3}$$

where $[\cdot \| \cdot]$ denotes concatenation. We perform self-attention on the new $\boldsymbol{z}^{(i)}$ for both the patch embedding $\boldsymbol{x}^{(i)}$ and the router embedding $\boldsymbol{u}^{(i)}$:

$$\begin{aligned}
\boldsymbol{x}^{(i)} &\leftarrow \text{Attn}^{\text{Intra}}\left(\boldsymbol{x}^{(i)}, \boldsymbol{z}^{(i)}, \boldsymbol{z}^{(i)}\right) \\
\boldsymbol{u}^{(i)} &\leftarrow \text{Attn}^{\text{Intra}}\left(\boldsymbol{u}^{(i)}, \boldsymbol{z}^{(i)}, \boldsymbol{z}^{(i)}\right)
\end{aligned} \tag{4}$$

where $\mathrm{Attn}\left(\boldsymbol{Q}, \boldsymbol{K}, \boldsymbol{V}\right)$ denotes the scaled dot-product self-attention mechanism in [30]. Note that the router embedding $\boldsymbol{u}^{(i)}$ is updated concurrently with the patch embedding $\boldsymbol{x}^{(i)}$ to maintain consistency, ensuring that the router effectively summarizes each granularity's features in the current training step and is ready for the subsequent inter-granularity self-attention. The intra-granularity self-attention mechanism allows the model to capture temporal features within a single granularity, facilitating the extraction of local features and correlations among timestamps at the same scale.

**b) Inter-Granularity Self-Attention.** We concatenate all router embeddings $\left\{\boldsymbol{u}^{(1)}, \boldsymbol{u}^{(2)}, \ldots, \boldsymbol{u}^{(n)}\right\}$ to form a sequence of routers $\boldsymbol{U} \in \mathbb{R}^{n \times D}$:

$$\boldsymbol{U} = \left[\boldsymbol{u}^{(1)} \| \boldsymbol{u}^{(2)} \| \ldots \| \boldsymbol{u}^{(n)}\right] \tag{5}$$

where $n$ is the number of different granularities. For granularity $i$ with patch length $L_i$, we apply self-attention to the router embedding $\boldsymbol{u}^{(i)} \in \mathbb{R}^{1 \times D}$ with all the routers $\boldsymbol{U}$:

$$\boldsymbol{u}^{(i)} \leftarrow \mathrm{Attn}^{\mathrm{Inter}}\left(\boldsymbol{u}^{(i)}, \boldsymbol{U}, \boldsymbol{U}\right) \tag{6}$$

Each router contains global information specific to one granularity by doing intra-granularity self-attention. By performing self-attention among routers, information can be exchanged and learned across different granularities, effectively capturing features across various scales. Additionally, the use of the router mechanism successfully reduces the time complexity of the naive approach from $O\left(\left(\sum_{i=1}^{n} N_i\right)^2\right)$ to $O\left(\sum_{i=1}^{n} N_i^2 + n^2\right)$. Given that $N_i \leq T$, the worst-case time complexity for our self-attention mechanism is $O\left(nT^2 + n^2\right)$. However, a reasonable choice of patch lengths as a power series, i.e., $L_i = 2^i$, leads to a time complexity of $O(T^2)$. To further reduce complexity and memory consumption, we apply shared layer normalization and feed-forward layers across all granularities. See appendix F for more details about complexity analysis.

**Summary.** Our method utilizes the standard transformer architecture shown in Figure 3. For given sample $\boldsymbol{x}_{\mathrm{in}}$, after $M$ layers of self-attention learning, we obtain a list of updated patch embeddings $\left\{\boldsymbol{x}^{(1)}, \boldsymbol{x}^{(2)}, \ldots, \boldsymbol{x}^{(n)}\right\}$, which we concatenate them to form a final representation $\boldsymbol{h}$ that can be used to predict label $y \in \mathbb{R}^K$ in a downstream classification task. Note that although we discuss multi-granularity here, our method is flexible and can be easily adapted to variants such as single-granularity or even repetitive same granularities. See Appendix D.2 for more details.

## 5 Experiments

We compare our Medformer with 10 baselines across 5 datasets, including 3 EEG datasets and 2 ECG datasets. Our method is evaluated under two setups (Section 3): subject-dependent and subject-independent. In the subject-dependent setup, training, validation, and test sets are split based on samples, while in the subject-independent setup, they are split based on subjects.

**Table 2: The information of processed datasets.** The table shows the number of subjects, samples, classes, channels, sampling rate, sample timestamps, modality of MedTS, and file size. Here, **#-Timestamps** indicates the number of timestamps per sample.

| Datasets | #-Subject | #-Sample | #-Class | #-Channel | #-Timestamps | Sampling Rate | Modality | File Size |
|---|---|---|---|---|---|---|---|---|
| APAVA | 23 | 5,967 | 2 | 16 | 256 | 256Hz | EEG | 186MB |
| ADFTD | 88 | 69,752 | 3 | 19 | 256 | 256Hz | EEG | 2.52GB |
| TDBrain | 72 | 6,240 | 2 | 33 | 256 | 256Hz | EEG | 571MB |
| PTB | 198 | 64,356 | 2 | 15 | 300 | 250Hz | ECG | 2.15GB |
| PTB-XL | 17,596 | 191,400 | 5 | 12 | 250 | 250Hz | ECG | 4.28GB |

**Datasets.** (1) **APAVA** [67] is an EEG dataset where each sample is assigned a binary label indicating whether the subject has Alzheimer's disease. (2) **TDBRAIN** [68] is an EEG dataset with a binary label assigned to each sample, indicating whether the subject has Parkinson's disease. (3) **ADFTD** [69, 19] is an EEG dataset with a three-class label for each sample, categorizing the subject as Healthy, having Frontotemporal Dementia, or Alzheimer's disease. (4) **PTB** [70] is an ECG dataset where each sample

**Table 3: Results of Subject-Dependent Setup.** The training, validation, and test sets are split based on samples according to a predetermined ratio. Results of the ADFTD dataset under this setup are presented here.

| Datasets | Models | Accuracy | Precision | Recall | F1 score | AUROC | AUPRC |
|---|---|---|---|---|---|---|---|
| **ADFTD** (3-Classes) | **Autoformer** | $87.83_{\pm1.62}$ | $87.63_{\pm1.66}$ | $87.22_{\pm1.97}$ | $87.38_{\pm1.79}$ | $96.59_{\pm0.88}$ | $93.82_{\pm1.64}$ |
| | **Crossformer** | $89.35_{\pm1.32}$ | $89.00_{\pm1.44}$ | $88.79_{\pm1.37}$ | $88.88_{\pm1.40}$ | $97.52_{\pm0.58}$ | $95.45_{\pm1.03}$ |
| | **FEDformer** | $77.63_{\pm2.37}$ | $76.76_{\pm2.17}$ | $76.68_{\pm2.48}$ | $76.60_{\pm2.46}$ | $91.67_{\pm1.34}$ | $84.94_{\pm2.11}$ |
| | **Informer** | $90.93_{\pm0.90}$ | $90.74_{\pm0.71}$ | $90.50_{\pm1.14}$ | $90.60_{\pm0.94}$ | $98.19_{\pm0.27}$ | $96.51_{\pm0.49}$ |
| | **iTransformer** | $64.90_{\pm0.25}$ | $62.53_{\pm0.27}$ | $62.21_{\pm0.26}$ | $62.25_{\pm0.33}$ | $81.52_{\pm0.29}$ | $68.87_{\pm0.49}$ |
| | **MTST** | $65.08_{\pm0.69}$ | $63.85_{\pm0.80}$ | $62.71_{\pm0.64}$ | $63.03_{\pm0.58}$ | $81.36_{\pm0.56}$ | $69.34_{\pm0.89}$ |
| | **Nonformer** | $96.12_{\pm0.47}$ | $95.94_{\pm0.56}$ | $95.99_{\pm0.38}$ | $95.96_{\pm0.47}$ | $99.59_{\pm0.09}$ | $99.08_{\pm0.16}$ |
| | **PatchTST** | $66.26_{\pm0.40}$ | $65.08_{\pm0.41}$ | $64.97_{\pm0.51}$ | $64.95_{\pm0.42}$ | $83.07_{\pm0.45}$ | $71.70_{\pm0.61}$ |
| | **Reformer** | $91.51_{\pm1.75}$ | $91.15_{\pm1.79}$ | $91.65_{\pm1.56}$ | $91.14_{\pm1.83}$ | $98.85_{\pm0.35}$ | $97.88_{\pm0.60}$ |
| | **Transformer** | $97.00_{\pm0.43}$ | $96.87_{\pm0.53}$ | $96.86_{\pm0.36}$ | $96.86_{\pm0.44}$ | $99.75_{\pm0.04}$ | $99.42_{\pm0.07}$ |
| | **Medformer (Ours)** | $\mathbf{97.62_{\pm0.34}}$ | $\mathbf{97.53_{\pm0.33}}$ | $\mathbf{97.48_{\pm0.40}}$ | $\mathbf{97.50_{\pm0.36}}$ | $\mathbf{99.83_{\pm0.05}}$ | $\mathbf{99.62_{\pm0.12}}$ |

is labeled with a binary indicator of Myocardial Infarction. (5) **PTB-XL** [71] is an ECG dataset with a five-class label for each sample, representing various heart conditions. Table 2 provides information on the processed datasets. For additional details on data characteristics, train-validation-test splits under different setups, and data preprocessing, please see Appendix B.

**Baselines.** We compare with 10 state-of-the-art time series transformer methods: Autoformer [28], Crossformer [33], FEDformer [52], Informer [29], iTransformer [31], MTST [53], Nonformer [54], PatchTST [32], Reformer [56], and vanilla Transformer [30].

**Implementation.** We employ six evaluation metrics: accuracy, precision (macro-averaged), recall (macro-averaged), F1 score (macro-averaged), AUROC (macro-averaged), and AUPRC (macro-averaged). The training process is conducted with five random seeds (41-45) on fixed training, validation, and test sets to compute the mean and standard deviation of the models. All experiments are run on an NVIDIA RTX 4090 GPU and a server with 4 RTX A5000 GPUs.

For data augmentation methods, we provide six widely used methods in time series augmentation [72, 66, 73, 61]. For more details about these six methods, see Appendix A. For the parameter tuning in our method and all baselines, we employ 6 layers for the encoder, set the dimension $D$ to 128, and the hidden dimension of feed-forward networks to 256. We utilize the Adam optimizer with a learning rate of 1e-4. The batch size is set to {32,32,128,128,128} for datasets APAVA, TDBrain, ADFTD, PTB, and PTB-XL, respectively. The training epoch is set to 100, with early stopping triggered after 10 epochs without improvement in the F1 score on the validation set. We save the model with the best F1 score on the validation set and evaluate it on the test set. See Appendix C for any additional implementation details of our method and all baselines.

## 5.1 Results of Subject-Dependent

**Setup.** In this setup, the training, validation, and test sets are split based on samples. All samples from all subjects are randomly shuffled and distributed into the training, validation, and test sets according to a predetermined ratio, allowing samples from the same subject to appear in three sets simultaneously. As discussed in the Preliminaries section 3, this setup has limited applicability for MedTS-based disease diagnosis in real-world scenarios. It is usually used to evaluate whether the dataset exhibits cross-subject features quickly. The results obtained from this setup are typically much higher than those from the subject-independent setup, showing a dataset's upper limit of learnability.

**Results.** We evaluate the EEG dataset ADFTD using this setup to provide a direct comparison of results with the subject-independent setup. The results are presented in Table 3. Our method outperforms all the baselines, achieving the top-1 results in all six evaluations, with an impressive F1 score of 97.50%. Notably, baseline methods like Informer, Nonformer, Reformer, and Transformer also demonstrate strong performance, achieving F1 scores exceeding 90%. The overall results indicate the presence of discernible and learnable features related to Alzheimer's Disease within this dataset.

## 5.2 Results of Subject-Independent

**Setup.** In this setup, the training, validation, and test sets are split based on subjects. All subjects and their corresponding samples are distributed into the training, validation, and test sets according to a predetermined ratio or subject IDs. Samples from the same subjects should exclusively appear in one

**Table 4: Results of Subject-Independent Setup.** The training, validation, and test sets are distributed based on subjects according to a predetermined ratio/IDs. Results of the APAVA, TDBrain, ADFTD, PTB, and PTB-XL datasets under this setup are presented here.

| Datasets | Models | Accuracy | Precision | Recall | F1 score | AUROC | AUPRC |
|---|---|---|---|---|---|---|---|
| APAVA (2-Classes) | Autoformer | 68.64±1.82 | 68.48±2.10 | 68.77±2.27 | 68.06±1.94 | 75.94±3.61 | 74.38±4.05 |
| | Crossformer | 73.77±1.95 | 79.29±4.36 | 68.86±1.70 | 68.93±1.85 | 72.39±3.33 | 72.05±3.65 |
| | FEDformer | 74.94±2.15 | 74.59±1.50 | 73.56±3.55 | 73.51±3.39 | 83.72±1.97 | 82.94±2.37 |
| | Informer | 73.11±4.40 | 75.17±6.06 | 69.17±4.56 | 69.47±5.06 | 70.46±4.91 | 70.75±5.27 |
| | iTransformer | 74.55±1.66 | 74.77±2.10 | 71.76±1.72 | 72.30±1.79 | **85.59±1.55** | **84.39±1.57** |
| | MTST | 71.14±1.59 | 79.30±0.97 | 65.27±2.28 | 64.01±3.16 | 68.87±2.34 | 71.06±1.60 |
| | Nonformer | 71.89±3.81 | 71.80±4.58 | 69.44±3.56 | 69.74±3.84 | 70.55±2.96 | 70.78±4.08 |
| | PatchTST | 67.03±1.65 | 78.76±1.28 | 59.91±2.02 | 55.97±3.10 | 65.65±0.28 | 67.99±0.76 |
| | Reformer | 78.70±2.00 | **82.50±3.95** | 75.00±1.61 | 75.93±1.82 | 73.94±1.40 | 76.04±1.14 |
| | Transformer | 76.30±4.72 | 77.64±5.95 | 73.09±5.01 | 73.75±5.38 | 72.50±6.60 | 73.23±7.60 |
| | Medformer (Ours) | **78.74±0.64** | 81.11±0.84 | **75.40±0.66** | **76.31±0.71** | 83.20±0.91 | 83.66±0.92 |
| TDBrain (2-Classes) | Autoformer | 87.33±3.79 | 88.06±3.56 | 87.33±3.79 | 87.26±3.84 | 93.81±2.26 | 93.32±2.42 |
| | Crossformer | 81.56±2.19 | 81.97±2.25 | 81.56±2.19 | 81.50±2.20 | 91.20±1.78 | 91.51±1.71 |
| | FEDformer | 78.13±1.98 | 78.52±1.91 | 78.13±1.98 | 78.04±2.01 | 86.56±1.86 | 86.48±1.99 |
| | Informer | 89.02±2.50 | 89.43±2.14 | 89.02±2.50 | 88.98±2.54 | 96.64±0.68 | 96.75±0.63 |
| | iTransformer | 74.67±1.06 | 74.71±1.06 | 74.67±1.06 | 74.65±1.06 | 83.37±1.14 | 83.73±1.27 |
| | MTST | 76.96±3.76 | 77.24±3.59 | 76.96±3.76 | 76.88±3.83 | 85.27±4.46 | 82.81±5.64 |
| | Nonformer | 87.88±2.48 | 88.86±1.84 | 87.88±2.48 | 87.78±2.56 | **97.05±0.68** | **96.99±0.68** |
| | PatchTST | 79.25±3.79 | 79.60±4.09 | 79.25±3.79 | 79.20±3.77 | 87.95±4.96 | 86.36±6.67 |
| | Reformer | 87.92±2.01 | 88.64±1.40 | 87.92±2.01 | 87.85±2.08 | 96.30±0.54 | 96.40±0.45 |
| | Transformer | 87.17±1.67 | 87.99±1.68 | 87.17±1.67 | 87.10±1.68 | 96.28±0.92 | 96.34±0.81 |
| | Medformer (Ours) | **89.62±0.81** | **89.68±0.78** | **89.62±0.81** | **89.62±0.81** | 96.41±0.35 | 96.51±0.33 |
| ADFTD (3-Classes) | Autoformer | 45.25±1.48 | 43.67±1.94 | 42.96±2.03 | 42.59±1.85 | 61.02±1.82 | 43.10±2.30 |
| | Crossformer | 50.45±2.31 | 45.57±1.63 | 45.88±1.82 | 45.50±1.70 | 66.45±2.03 | 48.33±2.05 |
| | FEDformer | 46.30±0.59 | 46.05±0.76 | 44.22±1.38 | 43.91±1.37 | 62.62±1.75 | 46.11±1.44 |
| | Informer | 48.45±1.96 | 46.54±1.68 | 46.06±1.84 | 45.74±1.38 | 65.87±1.27 | 47.60±1.30 |
| | iTransformer | 52.60±1.59 | 46.79±1.27 | 47.28±1.29 | 46.79±1.13 | 67.26±1.16 | 49.53±1.21 |
| | MTST | 45.60±2.03 | 44.70±1.33 | 45.05±1.30 | 44.31±1.74 | 62.50±0.81 | 45.16±0.85 |
| | Nonformer | 49.95±1.05 | 47.71±0.97 | 47.46±1.50 | 46.96±1.35 | 66.23±1.37 | 47.33±1.78 |
| | PatchTST | 44.37±0.95 | 42.40±1.13 | 42.06±1.48 | 41.97±1.37 | 60.08±1.50 | 42.49±1.79 |
| | Reformer | 50.78±1.17 | 49.64±1.49 | 49.89±1.67 | 47.94±0.69 | 69.17±1.58 | **51.73±1.94** |
| | Transformer | 50.47±2.14 | 49.13±1.83 | 48.01±1.53 | 48.09±1.59 | 67.93±1.59 | 48.93±2.02 |
| | Medformer (Ours) | **53.27±1.54** | **51.02±1.57** | **50.71±1.55** | **50.65±1.51** | **70.93±1.19** | 51.21±1.32 |
| PTB (2-Classes) | Autoformer | 73.35±2.10 | 72.11±2.89 | 63.24±3.17 | 63.69±3.84 | 78.54±3.48 | 74.25±3.53 |
| | Crossformer | 80.17±3.79 | 85.04±1.83 | 71.25±6.29 | 72.75±7.19 | 88.55±3.45 | 87.31±3.25 |
| | FEDformer | 76.05±2.54 | 77.58±3.61 | 66.10±3.55 | 67.14±4.37 | 85.93±4.31 | 82.59±5.42 |
| | Informer | 78.69±1.68 | 82.87±1.02 | 69.19±2.90 | 70.84±3.47 | 92.09±0.53 | 90.02±0.60 |
| | iTransformer | **83.89±0.71** | **88.25±1.18** | 76.39±1.01 | 79.06±1.06 | 91.18±1.16 | **90.93±0.98** |
| | MTST | 76.59±1.90 | 79.88±1.90 | 66.31±2.95 | 67.38±3.71 | 86.86±2.75 | 83.75±2.84 |
| | Nonformer | 78.66±0.49 | 82.77±0.86 | 69.12±0.87 | 70.90±1.00 | 89.37±2.51 | 86.67±2.38 |
| | PatchTST | 74.74±1.62 | 76.94±1.51 | 63.89±2.71 | 64.36±3.38 | 88.79±0.91 | 83.39±0.96 |
| | Reformer | 77.96±2.13 | 81.72±1.61 | 68.20±3.35 | 69.65±3.88 | 91.13±0.74 | 88.42±1.30 |
| | Transformer | 77.37±1.02 | 81.84±0.66 | 67.14±1.80 | 68.47±2.19 | 90.08±1.76 | 87.22±1.68 |
| | Medformer (Ours) | 83.50±2.01 | 85.19±0.94 | **77.11±3.39** | **79.18±3.31** | **92.81±1.48** | 90.32±1.54 |
| PTB-XL (5-Classes) | Autoformer | 61.68±2.72 | 51.60±1.64 | 49.10±1.52 | 48.85±2.27 | 82.04±1.44 | 51.93±1.71 |
| | Crossformer | **73.30±0.14** | 65.06±0.35 | **61.23±0.33** | 62.59±0.14 | **90.02±0.06** | **67.43±0.22** |
| | FEDformer | 57.20±9.47 | 52.38±6.09 | 49.04±7.26 | 47.89±8.44 | 82.13±4.17 | 52.31±7.03 |
| | Informer | 71.43±0.32 | 62.64±0.60 | 59.12±0.47 | 60.44±0.43 | 88.65±0.09 | 64.76±0.17 |
| | iTransformer | 69.28±0.22 | 59.59±0.45 | 54.62±0.18 | 56.20±0.19 | 86.71±0.10 | 60.27±0.21 |
| | MTST | 72.14±0.27 | 63.84±0.72 | 60.01±0.81 | 61.43±0.38 | 88.97±0.33 | 65.83±0.51 |
| | Nonformer | 70.56±0.55 | 61.57±0.66 | 57.75±0.72 | 59.10±0.66 | 88.32±0.36 | 63.40±0.79 |
| | PatchTST | 73.23±0.25 | **65.70±0.64** | 60.82±0.76 | **62.61±0.34** | 89.74±0.19 | 67.32±0.22 |
| | Reformer | 71.72±0.43 | 63.12±1.02 | 59.20±0.75 | 60.69±0.18 | 88.80±0.24 | 64.72±0.47 |
| | Transformer | 70.59±0.44 | 61.57±0.65 | 57.62±0.35 | 59.05±0.25 | 88.21±0.16 | 63.36±0.29 |
| | Medformer (Ours) | 72.87±0.23 | 64.14±0.42 | 60.60±0.46 | 62.02±0.37 | 89.66±0.13 | 66.39±0.22 |

of these three sets. This setup simulates real-world MedTS-based disease diagnosis, aiming to train a model on subjects with known labels and then test it on unseen subjects to determine if they have a specific disease. The challenges associated with this setup are discussed in section 3. All five datasets are evaluated using this setup.

**Results.** Table 4 presents the results of the subject-independent setup. Our method achieves the top-1 F1 scores on 4 out of 5 datasets. Overall, our method achieves 15 top-1 and 30 top-3 rankings out of 30 evaluations conducted across 5 datasets and 10 baselines, considering 6 different metrics. Figure 4 provides an overview heatmap table of average rank across 5 datasets on 6 metrics for all methods. Lower rank numbers indicate better results, with rank 1 representing the best performance among all methods. Our method demonstrates the best average rank among all methods across the 6 metrics. Additionally, it is notable that the result for ADFTD is a 50.65% F1 score under the subject-independent setup, which is significantly lower than the 97.50% F1 score achieved under the subject-dependent setup. This comparison highlights the challenge of the subject-independent setup.

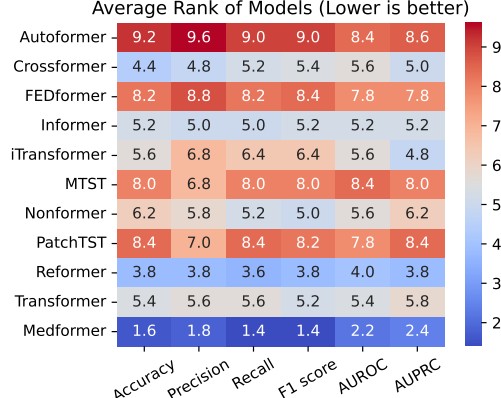

Figure 4: **Average Rank of Subject-Independent Setup.** The heatmap table shows the average rank of Medformer and 10 baselines across 5 datasets using the subject-independent setup. A lower number indicates better results. The average rank is calculated across the 5 datasets to obtain the overall average rank.

### 5.3 Ablation Study and Additional Experiments

**Ablation Study.** 1) *Module Study*: We conduct a module study to evaluate each proposed mechanism in our method (Appendix D.1). 2) *Patch Length Study*: We perform parameter tuning on the patch lengths to evaluate the effectiveness of multi-granularities (Appendix D.2). **Additional Experiments.** We also perform experiments on two human activities recognition datasets [74, 75] to demonstrate the learning ability of our model on general time series with potential channel correlations (Appendix E).

## 6 Conclusion and Limitations

**Conclusion** This paper presents Medformer, a multi-granularity patching transformer tailored for MedTS classification. We introduce three novel mechanisms that leverage the distinctive features of MedTS. These mechanisms include cross-channel patching to capture multi-timestamp and cross-channel features, multi-granularity embedding to learn features at various scales, and a two-stage multi-granularity self-attention mechanism to extract features both within and across granularities. Experimental results across five datasets, evaluated against ten baselines under the subject-independent setup, demonstrate the effectiveness of our method, showing its potential for real-world applications.

**Limitations and Future Work** The design of Medformer allows for inputting various patch lengths, offering opportunities and challenges. While varying patch lengths have been shown to outperform uniform lengths in many cases (see Appendix D.2 and Appendix C), not all patch length combinations yield optimal results. Some combinations may perform worse than uniform patch lengths, necessitating careful tuning of patch lengths. Future work could explore mechanisms for automatically selecting patch lengths. Additionally, developing modules that decompose subject-specific features from task-related features could further enhance learning under the subject-independent setup, presenting an intriguing direction for future research.

## Acknowledgments and Disclosure of Funding

This work is partially supported by the National Science Foundation under Grant No. 2245894. Any opinions, findings, conclusions or recommendations expressed in this material are those of the authors and do not necessarily reflect the views of the funders.

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

# Appendix A  Data Augmentation Banks

In the embedding stage, we apply data augmentation to the patch embeddings. We utilize a bank of data augmentation techniques to enhance the model's robustness and generalization. During the forward pass in training, each patch will pick one augmentation from available augmentation options with equal probability. The data augmentation methods include temporal flipping, channel shuffling, temporal masking, frequency masking, jittering, and dropout, and can be further expanded to more choices. We provide a detailed description of each technique below.

**Temporal Flippling** We reverse the MedTS data along the temporal dimension. The probability of applying this augmentation is controlled by a parameter *prob*, with a default value of 0.5.

**Channel Shuffling** We randomly shuffle the order of MedTS channels. The probability of applying channel shuffling is controlled by the parameter *prob*, also set by default to 0.5.

**Temporal Masking** We randomly mask some timestamps across all channels. The proportion of timestamps masked is controlled by the parameter *ratio*, with a default value of 0.1.

**Frequency Masking** First introduced in [73] for contrastive learning, this method involves converting the MedTS data into the frequency domain, randomly masking some frequency bands, and then converting it back. The proportion of frequency bands masked is controlled by the parameter *ratio*, with a default value of 0.1.

**Jittering** Random noise, ranging from 0 to 1, is added to the raw data. The intensity of the noise is adjusted by the parameter *scale*, which is set by default to 0.1.

**Dropout** Similar to the dropout layer in neural networks, this method randomly drops some values. The proportion of values dropped is controlled by the parameter *ratio*, with a default setting of 0.1.

# Appendix B  Data Preprocessing

## B.1  APAVA Preprocessing

The **A**lzheimer's **P**atients' Relatives **A**ssociation of **Va**lladolid (APAVA) dataset[*], referenced in the paper [67], is a public EEG time series dataset with 2 classes and 23 subjects, including 12 Alzheimer's disease patients and 11 healthy control subjects. On average, each subject has $30.0 \pm 12.5$ trials, with each trial being a 5-second time sequence consisting of 1280 timestamps across 16 channels. Before further preprocessing, each trial is scaled using the standard scaler. Subsequently, we segment each trial into 9 half-overlapping samples, where each sample is a 1-second time sequence comprising 256 timestamps. This process results in 5,967 samples. Each sample has a subject ID to indicate its originating subject. For the training, validation, and test set splits, we employ the subject-independent setup. Samples with subject IDs {15,16,19,20} and {1,2,17,18} are assigned to the validation and test sets, respectively. The remaining samples are allocated to the training set.

## B.2  TDBrain Preprocessing

The TDBrain dataset[*], referenced in the paper [68], is a large permission-accessible EEG time series dataset recording brain activities of 1274 subjects with 33 channels. Each subject has two trials: one under eye open and one under eye closed setup. The dataset includes a total of 60 labels, with each subject potentially having multiple labels indicating multiple diseases simultaneously. In this paper, we utilize a subset of this dataset containing 25 subjects with Parkinson's disease and 25 healthy controls, all under the eye-closed task condition. Each eye-closed trial is segmented into non-overlapping 1-second samples with 256 timestamps, and any samples shorter than 1-second are discarded. This process results in 6,240 samples. Each sample is assigned a subject ID to indicate its originating subject. For the training, validation, and test set splits, we employ the subject-independent setup. Samples with subject IDs {18,19,20,21,46,47,48,49} are assigned to the validation set, while samples with subject IDs {22,23,24,25,50,51,52,53} are assigned to the test set. The remaining samples are allocated to the training set.

---

[*]https://osf.io/jbysn/
[*]https://brainclinics.com/resources/

## B.3 ADFTD Preprocessing

The **A**lzheimer's **D**isease and **F**ron**T**otemporal **D**ementia (ADFTD) dataset[*], referenced in the papers [69, 19], is a public EEG time series dataset with 3 classes, including 36 Alzheimer's disease (AD) patients, 23 Frontotemporal Dementia (FTD) patients, and 29 healthy control (HC) subjects. The dataset has 19 channels, and the raw sampling rate is 500Hz. Each subject has a trial, with trial durations of approximately 13.5 minutes for AD subjects (min=5.1, max=21.3), 12 minutes for FD subjects (min=7.9, max=16.9), and 13.8 minutes for HC subjects (min=12.5, max=16.5). A bandpass filter between 0.5-45Hz is applied to each trial. We downsample each trial to 256Hz and segment them into non-overlapping 1-second samples with 256 timestamps, discarding any samples shorter than 1 second. This process results in 69,752 samples. For the training, validation, and test set splits, we employ both the subject-dependent and subject-independent setups. For the subject-dependent setup, we allocate 60%, 20%, and 20% of total samples into the training, validation, and test sets, respectively. For the subject-independent setup, we allocate 60%, 20%, and 20% of total subjects with their corresponding samples into the training, validation, and test sets, respectively.

## B.4 PTB Preprocessing

The PTB dataset[*], referenced in the paper [70], is a public ECG time series recording from 290 subjects, with 15 channels and a total of 8 labels representing 7 heart diseases and 1 health control. The raw sampling rate is 1000Hz. For this paper, we utilize a subset of 198 subjects, including patients with Myocardial infarction and healthy control subjects. We first downsample the sampling frequency to 250Hz and normalize the ECG signals using standard scalers. Subsequently, we process the data into single heartbeats through several steps. We identify the R-Peak intervals across all channels and remove any outliers. Each heartbeat is then sampled from its R-Peak position, and we ensure all samples have the same length by applying zero padding to shorter samples, with the maximum duration across all channels serving as the reference. This process results in 64,356 samples. For the training, validation, and test set splits, we employ the subject-independent setup. Specifically, we allocate 60%, 20%, and 20% of the total subjects, along with their corresponding samples, into the training, validation, and test sets, respectively.

## B.5 PTB-XL Preprocessing

The PTB-XL dataset[*], referenced in the paper [71], is a large public ECG time series dataset recorded from 18,869 subjects, with 12 channels and 5 labels representing 4 heart diseases and 1 healthy control category. Each subject may have one or more trials. To ensure consistency, we discard subjects with varying diagnosis results across different trials, resulting in 17,596 subjects remaining. The raw trials consist of 10-second time intervals, with sampling frequencies of 100Hz and 500Hz versions. For our paper, we utilize the 500Hz version, then we downsample to 250Hz and normalize using standard scalers. Subsequently, each trial is segmented into non-overlapping 1-second samples with 250 timestamps, discarding any samples shorter than 1 second. This process results in 191,400 samples. For the training, validation, and test set splits, we employ the subject-independent setup. Specifically, we allocate 60%, 20%, and 20% of the total subjects, along with their corresponding samples, into the training, validation, and test sets, respectively.

## Appendix C   Implementation Details

We implement our method and all the baselines based on the Time-Series-Library project[*] from Tsinghua University [76], which integrates all methods under the same framework and training techniques to ensure a relatively fair comparison. The 10 baseline time series transformer methods are Autoformer [28], Crossformer [33], FEDformer [52], Informer [29], iTransformer [31], MTST [53], Nonformer [54], PatchTST [32], Reformer [56], and vanilla Transformer [30].

---

[*]https://openneuro.org/datasets/ds004504/versions/1.0.6
[*]https://physionet.org/content/ptbdb/1.0.0/
[*]https://physionet.org/content/ptb-xl/1.0.3/
[*]https://github.com/thuml/Time-Series-Library

For all methods, we employ 6 layers for the encoder, with the self-attention dimension $D$ set to 128 and the hidden dimension of the feed-forward networks set to 256. The optimizer used is Adam, with a learning rate of 1e-4. The batch size is set to {32,32,128,128,128} for the datasets APAVA, TDBrain, ADFD, PTB, and PTB-XL, respectively. Training is conducted for 100 epochs, with early stopping triggered after 10 epochs without improvement in the F1 score on the validation set. We save the model with the best F1 score on the validation set and evaluate it on the test set. We employ six evaluation metrics: accuracy, precision (macro-averaged), recall (macro-averaged), F1 score (macro-averaged), AUROC (macro-averaged), and AUPRC (macro-averaged). Both subject-dependent and subject-independent setups are implemented for different datasets. Each experiment is run with 5 random seeds (41-45) and fixed training, validation, and test sets to compute the average results and standard deviations.

**Medformer (Our Method)** We use a list of patch lengths in patch embedding to generate patches with different granularities. Instead of flattening the patches and mapping them to dimension $D$ during patch embedding, we use a conv2d network to directly map patches into a 1-D representation with dimension $D$. These patch lengths can vary, including different numbers of patch lengths such as $\{2, 4, 8, 16\}$, repetitive numbers such as $\{8, 8, 8, 8\}$, or a mix of different and repetitive lengths such as $\{8, 8, 8, 16, 16, 16\}$. It is also possible to use only one patch length, such as $\{8\}$, which indicates a single granularity. The patch lists used for the datasets APAVA, TDBrain, ADFD, PTB, and PTB-XL are $\{2, 2, 2, 4, 4, 4, 16, 16, 16, 16, 32, 32, 32, 32, 32\}$, $\{8, 8, 8, 16, 16, 16\}$, $\{2, 4, 8, 8, 16, 16, 16, 16, 32, 32, 32, 32, 32, 32, 32, 32\}$, $\{2, 4, 8, 8, 16, 16, 16, 32, 32, 32, 32, 32\}$, and $\{2, 4, 8, 8, 16, 16, 16, 16, 32, 32, 32, 32, 32, 32, 32, 32\}$, respectively. The data augmentations are randomly chosen from a list of four possible options: none, jitter, scale, and mask. The number following each augmentation method indicates the degree of augmentation. Detailed descriptions of these methods can be found in Appendix A. The augmentation methods used for the datasets APAVA, TDBrain, ADFD, PTB, and PTB-XL are {none, drop0.35}, {none, drop0.25}, {drop0.5}, {drop0.5}, and {jitter0.2, scale0.2, drop0.5}, respectively.

**Autoformer** Autoformer [28] employs an auto-correlation mechanism to replace self-attention for time series forecasting. Additionally, they use a time series decomposition block to separate the time series into trend-cyclical and seasonal components for improved learning. The raw source code is available at https://github.com/thuml/Autoformer.

**Crossformer** Crossformer [33] designs a single-channel patching approach for token embedding. They utilize two-stage self-attention to leverage both temporal features and channel correlations. A router mechanism is proposed to reduce time and space complexity during the cross-dimension stage. The raw code is available at https://github.com/Thinklab-SJTU/Crossformer.

**FEDformer** FEDformer [52] leverages frequency domain information using the Fourier transform. They introduce frequency-enhanced blocks and frequency-enhanced attention, which are computed in the frequency domain. A novel time series decomposition method replaces the layer norm module in the transformer architecture to improve learning. The raw code is available at https://github.com/MAZiqing/FEDformer.

**Informer** Informer [29] is the first paper to employ a one-forward procedure instead of an autoregressive method in time series forecasting tasks. They introduce ProbSparse self-attention to reduce complexity and memory usage. The raw code is available at https://github.com/zhouhaoyi/Informer2020.

**iTransformer** iTransformer [31] questions the conventional approach of embedding attention tokens in time series forecasting tasks and proposes an inverted approach by embedding the whole series of channels into a token. They also invert the dimension of other transformer modules, such as the layer norm and feed-forward networks. The raw code is available at https://github.com/thuml/iTransformer.

**MTST** MTST [53] uses the same token embedding method as Crossformer and PatchTST. It highlights the importance of different patching lengths in forecasting tasks and designs a method that can take different sizes of patch tokens as input simultaneously. The raw code is available at https://github.com/networkslab/MTST.

**Nonformer** Nonformer [54] analyzes the impact of non-stationarity in time series forecasting tasks and its significant effect on results. They design a de-stationary attention module and incorporate normalization and denormalization steps before and after training to alleviate the over-stationarization problem. The raw code is available at https://github.com/thuml/Nonstationary_Transformers.

Table 5: Module Study.

| Datasets | Models | Accuracy | Precision | Recall | F1 score | AUROC | AUPRC |
|---|---|---|---|---|---|---|---|
| **APAVA** | **No Inter-Attention** | $76.90_{\pm1.50}$ | $78.08_{\pm2.12}$ | $73.87_{\pm1.48}$ | $74.59_{\pm1.58}$ | $80.29_{\pm3.75}$ | $81.32_{\pm3.37}$ |
| | **No Augmentation** | $75.21_{\pm2.94}$ | $76.69_{\pm3.41}$ | $71.72_{\pm3.22}$ | $72.30_{\pm3.46}$ | $77.05_{\pm5.22}$ | $78.15_{\pm5.42}$ |
| | **Single-Channel Patching** | $73.08_{\pm1.34}$ | $76.43_{\pm1.46}$ | $68.6_{\pm1.93}$ | $68.68_{\pm2.37}$ | $69.54_{\pm0.64}$ | $69.43_{\pm1.36}$ |
| | **Medformer** | $\mathbf{78.74_{\pm0.64}}$ | $\mathbf{81.11_{\pm0.84}}$ | $\mathbf{75.40_{\pm0.66}}$ | $\mathbf{76.31_{\pm0.71}}$ | $\mathbf{83.20_{\pm0.91}}$ | $\mathbf{83.66_{\pm0.92}}$ |
| **TDBrain** | **No Inter-Attention** | $88.17_{\pm0.72}$ | $88.27_{\pm0.72}$ | $88.17_{\pm0.72}$ | $88.16_{\pm0.72}$ | $96.06_{\pm0.40}$ | $96.18_{\pm0.39}$ |
| | **No Augmentation** | $88.56_{\pm0.66}$ | $88.67_{\pm0.61}$ | $88.56_{\pm0.66}$ | $88.55_{\pm0.66}$ | $96.11_{\pm0.39}$ | $96.20_{\pm0.39}$ |
| | **Single-Channel Patching** | $80.94_{\pm0.95}$ | $81.84_{\pm1.55}$ | $80.94_{\pm0.95}$ | $80.81_{\pm0.92}$ | $89.65_{\pm0.85}$ | $89.48_{\pm0.91}$ |
| | **Medformer** | $\mathbf{89.62_{\pm0.81}}$ | $\mathbf{89.68_{\pm0.78}}$ | $\mathbf{89.62_{\pm0.81}}$ | $\mathbf{89.62_{\pm0.81}}$ | $\mathbf{96.41_{\pm0.35}}$ | $\mathbf{96.51_{\pm0.33}}$ |
| **ADFD** | **No Inter-Attention** | $52.14_{\pm1.11}$ | $\mathbf{51.13_{\pm2.57}}$ | $46.15_{\pm0.86}$ | $45.59_{\pm1.18}$ | $67.99_{\pm1.77}$ | $49.68_{\pm2.05}$ |
| | **No Augmentation** | $49.99_{\pm6.86}$ | $48.21_{\pm6.16}$ | $44.88_{\pm4.59}$ | $44.07_{\pm4.75}$ | $65.03_{\pm6.12}$ | $47.11_{\pm5.64}$ |
| | **Single-Channel Patching** | $47.09_{\pm1.22}$ | $45.42_{\pm1.30}$ | $43.94_{\pm0.80}$ | $44.11_{\pm0.84}$ | $62.07_{\pm0.86}$ | $44.57_{\pm0.95}$ |
| | **Medformer** | $\mathbf{53.27_{\pm1.54}}$ | $51.02_{\pm1.57}$ | $\mathbf{50.71_{\pm1.55}}$ | $\mathbf{50.65_{\pm1.51}}$ | $\mathbf{70.93_{\pm1.19}}$ | $\mathbf{51.21_{\pm1.32}}$ |
| **PTB** | **No Inter-Attention** | $78.02_{\pm2.70}$ | $80.96_{\pm1.39}$ | $68.65_{\pm4.58}$ | $69.97_{\pm5.27}$ | $\mathbf{92.94_{\pm0.86}}$ | $90.19_{\pm1.12}$ |
| | **No Augmentation** | $77.64_{\pm1.65}$ | $81.03_{\pm1.60}$ | $67.88_{\pm2.61}$ | $69.31_{\pm3.22}$ | $92.19_{\pm0.71}$ | $89.37_{\pm0.96}$ |
| | **Single-Channel Patching** | $79.02_{\pm1.62}$ | $81.14_{\pm1.59}$ | $70.43_{\pm2.47}$ | $72.24_{\pm2.76}$ | $85.74_{\pm1.59}$ | $82.23_{\pm1.48}$ |
| | **Medformer** | $\mathbf{83.50_{\pm2.01}}$ | $\mathbf{85.19_{\pm0.94}}$ | $\mathbf{77.11_{\pm3.39}}$ | $\mathbf{79.18_{\pm3.31}}$ | $92.81_{\pm1.48}$ | $\mathbf{90.32_{\pm1.54}}$ |
| **PTB-XL** | **No Inter-Attention** | $72.51_{\pm0.16}$ | $63.61_{\pm0.28}$ | $59.75_{\pm0.30}$ | $61.25_{\pm0.22}$ | $89.48_{\pm0.08}$ | $65.74_{\pm0.26}$ |
| | **No Augmentation** | $72.68_{\pm0.19}$ | $63.99_{\pm0.62}$ | $59.73_{\pm0.41}$ | $61.26_{\pm0.34}$ | $89.49_{\pm0.05}$ | $66.00_{\pm0.22}$ |
| | **Single-Channel Patching** | $72.79_{\pm0.35}$ | $\mathbf{64.80_{\pm0.51}}$ | $59.57_{\pm0.44}$ | $61.43_{\pm0.38}$ | $88.97_{\pm0.19}$ | $65.91_{\pm0.34}$ |
| | **Medformer** | $\mathbf{72.87_{\pm0.23}}$ | $64.14_{\pm0.42}$ | $\mathbf{60.60_{\pm0.46}}$ | $\mathbf{62.02_{\pm0.37}}$ | $\mathbf{89.66_{\pm0.13}}$ | $\mathbf{66.39_{\pm0.22}}$ |

**PatchTST** PatchTST [32] embeds a sequence of single-channel timestamps as a patch token to replace the attention token used in the vanilla transformer. This approach enlarges the receptive field and enhances forecasting ability. The raw code is available at https://github.com/yuqinie98/PatchTST.

**Reformer** Reformer [56] replaces dot-product attention with locality-sensitive hashing. They also use a reversible residual layer instead of standard residuals. The raw code is available at https://github.com/lucidrains/reformer-pytorch.

**Transformer** Transformer [30], commonly known as the vanilla transformer, is introduced in the well-known paper "Attention is All You Need." It can also be applied to time series by embedding each timestamp of all channels as an attention token. The PyTorch version of the code is available at https://github.com/jadore801120/attention-is-all-you-need-pytorch.

# Appendix D    Ablation Study

## D.1    Module Study

To assess the efficacy of our proposed mechanisms—inter-granularity self-attention, embedding augmentation, and multi-channel patching—we conduct ablation studies on five datasets across three distinct settings: without inter-granularity attention, without embedding augmentation, and with single-channel patching. We maintain the other two modules intact in each setting and fix all hyperparameters as described in the implementation details C. Table 5 presents a comparison between our full Medformer model and these three variants. The complete Medformer model secures 28 top-1 and 30 top-2 rankings across 30 evaluations, demonstrating robust performance. We observe that each module significantly enhances performance: on average, across the datasets, inter-granularity attention contributes to a 3.64% improvement in F1 score, embedding augmentation leads to a 4.46% increase and multi-channel patching results in a 6.10% enhancement in F1 score. We find multi-channel patching particularly beneficial for results, especially in EEG data. Overall, these results underscore the critical role of each component in our design.

## D.2    Patch Length Study

To investigate the effects of multi-granularity and computational complexity, we conduct an empirical analysis using various patch lengths on the APAVA dataset. Table 6 presents the evaluation results for different combinations of patch lengths. Initially, we compare the performance of models using a single patch length against models using five identical patch lengths (e.g., $\{8\}$ vs $\{8, 8, 8, 8, 8\}$). Our findings indicate that using repetitive patch lengths generally enhances performance, except when

**Table 6: Patch Length Study**

| Datasets | Models | Accuracy | Precision | Recall | F1 score | AUROC | AUPRC |
|---|---|---|---|---|---|---|---|
| APAVA | {2} | 71.82 ±8.13 | 73.23 ±8.91 | 69.69 ±6.31 | 69.95 ±7.08 | 69.34 ±4.72 | 69.18 ±5.17 |
| | {4} | 75.72 ±3.73 | 78.15 ±5.84 | 72.14 ±3.39 | 72.83 ±3.64 | 72.75 ±4.76 | 73.84 ±5.08 |
| | {8} | 71.29 ±3.02 | 72.83 ±2.73 | 67.38 ±4.25 | 67.17 ±4.98 | 76.12 ±4.25 | 76.74 ±4.29 |
| | {12} | 69.77 ±4.01 | 69.72 ±5.65 | 67.09 ±3.34 | 67.36 ±3.53 | 75.19 ±3.00 | 75.36 ±3.48 |
| | {16} | 70.92 ±1.99 | 71.46 ±3.09 | 67.67 ±2.26 | 67.81 ±2.45 | 76.97 ±2.53 | 77.21 ±2.87 |
| | {24} | 71.68 ±2.44 | 74.26 ±3.49 | 67.14 ±2.55 | 67.13 ±2.85 | 79.07 ±3.34 | 78.73 ±3.32 |
| | {32} | 72.55 ±1.51 | 75.74 ±1.49 | 68.38 ±2.99 | 68.19 ±3.23 | 79.17 ±2.17 | 78.44 ±2.40 |
| | {2,2,2,2,2} | 65.52 ±8.24 | 65.97 ±7.41 | 64.14 ±6.06 | 63.71 ±7.23 | 63.15 ±3.43 | 61.84 ±4.81 |
| | {4,4,4,4,4} | 76.91 ±1.72 | 78.66 ±3.20 | 73.71 ±1.26 | 74.46 ±1.42 | 74.90 ±3.21 | 76.36 ±3.12 |
| | {8,8,8,8,8} | 71.81 ±3.81 | 74.25 ±6.34 | 67.72 ±3.45 | 67.89 ±3.68 | 74.95 ±5.34 | 75.59 ±5.63 |
| | {12,12,12,12,12} | 71.17 ±3.85 | 72.18 ±5.97 | 67.65 ±3.27 | 67.96 ±3.48 | 76.71 ±4.82 | 77.27 ±4.91 |
| | {16,16,16,16,16} | 71.13 ±3.33 | 72.14 ±5.69 | 67.82 ±2.45 | 68.13 ±2.58 | 76.34 ±4.52 | 76.39 ±4.94 |
| | {24,24,24,24,24} | 73.18 ±2.15 | 75.72 ±3.46 | 68.98 ±2.11 | 69.27 ±2.33 | 81.10 ±2.61 | 81.20 ±2.68 |
| | {32,32,32,32,32} | 74.34 ±2.20 | 78.92 ±1.57 | 69.66 ±2.80 | 69.80 ±3.42 | 81.11 ±1.10 | 80.69 ±1.02 |
| | {2,2,4,16,32} | **78.21** ±2.60 | **80.82** ±4.30 | **74.92** ±2.07 | **75.78** ±2.31 | **80.73** ±2.34 | **81.38** ±2.38 |

$L = 2$, suggesting that additional identical patch lengths can capture more information, analogous to multi-head attention mechanisms.

Furthermore, we assess the performance of a manually selected combination of varying patch lengths, specifically $\{2, 2, 4, 16, 32\}$. This configuration achieves the highest performance across all evaluated metrics, underscoring the effectiveness of our designed attention module in accommodating multi-granularity patches. However, it is worth noting that mixing different patch lengths does not guarantee improved performance. See G for more detailed discussion.

## Appendix E    Additional Experiments

**Table 7: Additional Datasets and Methods** We selected three old baselines in the previous experiments that showed strong performance: Crossformer, Reformer, and Transformer. Additionally, we introduce three new baselines: TCN, ModernTCN, and Mamba. These six baselines are evaluated on one old dataset in the previous experiments, TDBrain(6,240 samples, 2 classes), and two new HAR datasets: FLAAP (13,123 samples, 10 classes) and UCI-HAR (10,299 samples, 6 classes). The bold number denotes the best result, and the underlined number denotes the second best.

| Datasets | TDBrain (2 Classes) | | FLAAP (10 Classes) | | UCI-HAR (6 Classes) | |
|---|---|---|---|---|---|---|
| Metrics / Models | Accuracy | F1 Score | Accuracy | F1 Score | Accuracy | F1 Score |
| Crossformer | 81.56±2.19 | 81.50±2.20 | 75.84±0.52 | 75.52±0.66 | 89.74±1.08 | 89.70±1.10 |
| Reformer | 87.92±2.01 | 87.85±2.08 | 71.65±1.27 | 71.14±1.45 | 88.44±2.02 | 88.34±1.98 |
| Transformer | 87.17±1.67 | 87.10±1.68 | 74.96±1.25 | 74.49±1.39 | 88.86±1.65 | 88.80±1.67 |
| TCN | 80.92±2.94 | 80.82±3.03 | 66.48±1.66 | 65.29±1.74 | **93.08**±0.95 | **93.19**±0.88 |
| ModernTCN | 81.96±2.12 | 81.79±2.23 | 74.80±0.96 | 74.35±0.85 | 91.44±1.01 | 91.47±0.98 |
| Mamba | 89.58±0.74 | 89.58±0.73 | 64.87±2.78 | 64.14±2.70 | 87.78±1.10 | 87.72±1.10 |
| Medformer | **89.62**±0.81 | **89.62**±0.81 | **76.44**±0.64 | **76.25**±0.65 | 91.65±0.74 | 91.61±0.75 |

To evaluate the performance of our method on general time series, we test it on two human activity recognition (HAR) datasets: FLAAP [74] and UCI-HAR [75], which exhibit potential channel correlations inherently. Additionally, we compare our method with three other approaches: TCN [51], ModernTCN [16], and Mamba [77]. Our method achieves the highest top-1 accuracy and F1 score on TDBrain and FLAAP, and ranks second-best on UCI-HAR.

## Appendix F    Complexity Analysis

Let the number of timestamps $T$, and patch list $\{L_1, L_2, \ldots, L_n\}$ be given, where the $i$-th patch length $L_i$ produce $N_i = \lceil T/L_i \rceil$ number of patches. During intra-granularity attention, we perform self-attention among the patch embeddings within the same granularity. The total complexity is

$O\left(\sum_{i=1}^{n} N_i^2\right)$. During intra-granularity attention, we perform self-attention among $n$ routers, with a time complexity of $O(n^2)$. Therefore, the total time complexity is $O\left(n^2 + \sum_{i=1}^{n} N_i^2\right)$.

One potentially useful patch list is the power series $\{2^1, 2^2, \ldots 2^n\}$, where $2^n < T$. In this case, the complexity of intra-granularity attention reduces as follows:

$$
\begin{aligned}
O\left(\sum_{i=1}^{n} N_i^2\right) &= O\left(\sum_{i=1}^{n}\left\lceil\frac{T}{2^i}\right\rceil^2\right) \le O\left(\sum_{i=1}^{n}\left(\frac{T}{2^i}+1\right)^2\right) \\
&= O\left(\sum_{i=1}^{n}\left(\frac{T^2}{2^{2i}}+2\frac{T}{2^i}+1\right)\right) = O\left(T^2\sum_{i=1}^{n}\frac{1}{2^{2i}}+2T\sum_{i=1}^{n}\frac{1}{2^i}+n\right) \\
&\le O\left(\frac{1}{3}T^2+2T+\log T\right) = O(T^2)
\end{aligned}
$$

The complexity of inter-granularity attention is $O(n^2) \le O(\log^2 T)$. Therefore, the total time complexity of the two-stage multi-granularity self-attention module is $O(T^2)$, which is the same complexity as the vanilla transformer. This analysis demonstrates our model's ability to incorporate different granularities without significantly increasing computational overhead.

# Appendix G    Discussion

## G.1    Comparision with Other Multi-Granularity Methods

MTST [53] and Pathformer [55] differ from our Medformer in three significant aspects: (1) **Patching & Embedding** MTST and Pathformer utilize single-channel patching, presupposing channel independence. In contrast, Medformer employs multi-channel patching to capture potential channel correlations. (2) **Granularity Interactions** MTST assimilates multi-granularity information by concatenating outputs from different branches, while Pathformer uses adaptive pathways for weighted aggregation of these outputs without any inter-granularity interactions within the attention modules. In contrast, Medformer introduces a novel inter-granularity attention mechanism specifically designed for granularity interaction, thereby effectively integrating multi-granularity information.

Scaleformer [60] operates as a model-agnostic structural framework that employs variable downsampling and upsampling rates on embeddings outside of attention modules. Although it integrates seamlessly with non-patching methods like Autoformer and FEDformer, its incorporation into patching methods is not straightforward and may result in sub-optimal patch representations [53]. Consequently, the design objectives of Scaleformer are largely orthogonal to ours, which concentrate on multi-granularity patching and attention mechanisms.

# Appendix H    Broader Impacts

Our proposed model demonstrates performance comparable to or surpassing state-of-the-art baselines on medical time series classification tasks. The model's design, which includes specialized patching and self-attention mechanisms, specifically targets channel correlations and multi-granularity information. We anticipate our findings will encourage further research into effective strategies for capturing multi-scale information in medical time series data. Additionally, this work could broaden interest in medical time series classification, an area that remains less explored compared to time series forecasting.

Besides, different experiment setups based on medical perspectives, such as subject-dependent and subject-independent, are evaluated to simulate real-world applications. On a societal level, our model has potential applications in healthcare, such as facilitating the diagnosis of diseases using medical time series data. For instance, it could be employed to detect neurological disorders through EEG data. However, practitioners should be cognizant of the model's limitations.

