# OpenReview forum: "Medformer: A Multi-Granularity Patching Transformer for Medical Time-Series Classification"
_NeurIPS.cc/2024/Conference — NeurIPS 2024 poster_

### Official Review · Reviewer_ApjC · 2024-07-11

**Soundness:** 4
**Presentation:** 4
**Contribution:** 3
**Rating:** 7
**Confidence:** 3

**Summary:**

This paper introduces Medformer, a transformer variant designed to learn complex dynamics from medical signals. This is achieved in three key ways :
1. Cross-channel patching for token embedding;
2. Multi-length patching for coarse and fine feature processing;
3. Multi-granularity self-attention for information granularities integration;

The proposed model is then extensively validated through comparison to ten baselines on five datasets.

**Strengths:**

- Originality : The model proposed in this paper is the first to integrate both cross-channel patching with varying patch lengths as an embedding method, and a new multi-granularity self-attention mechanism.
- Quality : The performed tests are extensive (five different datasets, ten competing baselines). Reproducibility is largely facilitated as all datasets and the full code are publicly available. To validate all components, ablation studies are provided. Since the introduction of multi-granularity could seem like it would increase complexity, care has been put so that this is not the case (e.g. with the two-stage router-based self-attention).
- Clarity : The paper is well-written and easy to follow, and the figures are intelligible and informative. All equations and steps are clearly explained.
- Significance : Improving diagnosis performance is doubly useful in the medical domain, as it directly aids healthcare, and indirectly eases the adoption of models by clinicians. On the technical side, this paper opens the door for multi-granularity transformers able to learn from both fine and coarse-grained features.

**Weaknesses:**

This paper suffers from one main weakness : processing medical time series without providing explainability studies undermines its usability and transparency. This could be addressed in the future by adapting one of the techniques developed to interpret predictions made by transformer models.

The provided code could also benefit from more comments, for example in the `exp/*` files.

**Questions:**

- For clarification, does the random seed only affect the initialization of the model? Are the training/validation/test splits the same for all five runs?
- What was your reasoning behind the different augmentation techniques used depending on the dataset?

**Limitations:**

The authors have addressed the technical limitations of their work, albeit not in the main body of the article. However, they have not addressed the societal impact of their model, were it to be adopted in a healthcare setting : steps should be taken to either prevent or disclose clearly whether demographic biases (for example) found in the data are also found in the model's predictions.

---

> ### Author Rebuttal · Authors · 2024-08-07
>
> Thank you for your thoughtful feedback on our work！ We appreciate you carefully reading our method, equations, figures, and even the code! We are happy to receive your endorsement and feel that our paper is easy to follow! Again, thank you for your comments. Here are the responses to your questions and concerns.
>
> ***
>
> **Q1**: Processing medical time series should provide explanability and transparency.
>
> **A1**: Thanks for raising this concern. We agree explanability is crucial to medical tasks, therefore we further clarify how we can provide explanations from our trained model with an example from the TDBrain dataset. This is a dataset of EEG signals for prediction on Parkinson's disease, which is suitable for exploring if our model is utilizing the correct channels' information and the signals at the right frequency for a reasonable diagnosis. To find out important channels contributing to our model's prediction, we first mask out the rest of the brain regions and only use certain combinations of them to perform predictions. Important channels contribute most to the prediction accuracy, as shown by Fig R.1 (b) in the attached PDF. In Fig R.1 (b), we can find Central sulcus contributes most to the prediction accuracy, followed by Frontal, Occipital and Parietal lobes, and plural available regions are generally better than single region. This aligns with the central sulcus's role in movement and its close anatomy relationship with Basal Ganglia, which is the primary brain region indicating Parkinson's disease [9], meaning that our model has successfully learned to pick up informations from brain regions that matters much more. Furthermore, Fig R.1 (a) shows how different routers are attending to different segment of the same signal (deeper red means that the router attends more to this patch). This demonstrates that other than capturing features at the 32Hz and 16Hz scaffolded by the patch length $L=8$ and $L=16$ for 256Hz EEG signals, the model is as well focusing on brain waves at 8~10Hz due to which the model is "downsampling" the tokens with attention scores in stripe patterns. This precisely aligns with the interval at which the EEG signals differs the most between healthy individuals and patients [10] and therefore is crucial in the diagnosis process of Parkinson's disease. In conclusion, our method is capable of focusing on important channels as well as time-frequency information which is well-supported by literatures and may give inspiration for future studies and applications.
>
> ***
>
> **Q2**: The provided code could benefit from more comments.
>
> **A2**: Thanks for raising this concern. We will update and add more comments to our code in our next version.
>
> ***
>
> **Q3**: Does random seed only affect the initialization of model? Are the training/validation/test splits the same for all five runs?
>
> **A3**: Yes. The random seed only affects the model's initialization, while the train-test splits remain consistent across the 5 runs. To satisfy the subject-independent setup, where the train-test split are based on subjects, we fixed the subjects used in each set. This means that the subjects included in the training, validation, and test sets remain the same, and only the model's initialization is randomized for each run.
>
> ***
>
> **Q4**: What was your reasoning behind the different augmentation techniques used depending on the dataset?
>
> **A4**: We did some preliminary analysis and evaluations to choose augmentation methods. Our goal is to select augmentation methods that do not change the label for classification while introducing some variance in the samples. In general, timestamp masking is the safest meth for classification tasks, followed by jittering. Timestamp masking involves randomly masking certain timestamps in the time domain without damaging the semantic information. Jittering, when applied with a low ratio of random noise, also preserves the semantic information in the raw data. We also experimented with frequency band masking but found that it sometimes negatively impacted performance. We believe this could be due to the semantic information in some datasets being concentrated in specific frequency bands, which this type of augmentation may disrupt.
>
> ***
>
> **Q5**: Considering the societal impact of their model, steps should be taken to prevent the disclosing of demographic information in the dataset.
>
> **A5**: Thank you for raising this concern. During data preprocessing, we manually removed the demographic information in the dataset and did not incorporate it into model training. We strongly agree that protecting sensitive information in the data is important when dealing with medical data. To further protect the demongraphic information, we may incorporate techniques such as differential privacy in future works.

---

> > ### Comment · Reviewer_ApjC · 2024-08-08
> >
> > Thank you for your response. I appreciate your demonstration that Medformer can be explainable, and the clarifications about augmentation strategies and demographic information. I hope you will find a way to include this and your additional benchmarks in your final paper.
> >
> > My rating remains unchanged and positive (Accept); I do not think that a method being primarily applicable to clinical data and not general time series is grounds for rejection, since the "Machine learning for healthcare" area of NeurIPS exists for this purpose.

---

> > > ### Author Response · Authors · 2024-08-08
> > >
> > > Thank you again for your endorsement! We will include the explainable results and new experiments in our final paper.

---

### Official Review · Reviewer_zGJA · 2024-07-12

**Soundness:** 3
**Presentation:** 3
**Contribution:** 2
**Rating:** 6
**Confidence:** 3

**Summary:**

The authors have created a model that seems to perform especially well for high-frequency waveform classification. They have benchmarked their model over several datasets, and its overall rank is higher than that of other models.

**Strengths:**

The model successfully combines three (seemingly pre-existing ideas). The model seems to perform reasonably well in the provided benchmarks. The subject task distinction is an interesting addition and shows that thought has been put into the design of the tasks. The code is provided together with some documentation, significantly improving reproducibility.

**Weaknesses:**

The limitations and discussion should at least be briefly discussed in the paper itself, not just in the appendix. The compared models, while plentiful, seem only transformer-based; it would be interesting to see a comparison to different architectures, especially since TCN is also mentioned in the related work (e.g., try the Mamba model https://arxiv.org/pdf/2312.00752). The improvements over existing transformer models seem marginal.
The efficiency of the models should be benchmarked (e.g., training time).
Table 3: some results are tied as they lie in the range of the standard deviation of the best model. It would be good to highlight these results as well.
Line 724: Comparison misspelled.

**Questions:**

The authors use two types of vital signs to benchmark their model. Is there enough reason that we could assume the superior performance also translates to other vital signs and, more broadly, medical time-series prediction tasks?

**Limitations:**

Tasks could be extended to (cheap to collect) PPG signals. Additionally, one could think of a multi-model setup as we often have more (static or lower frequency time-series) data on specific test subjects.

---

> ### Author Rebuttal · Authors · 2024-08-07
>
> Thank you for your suggestions to our paper! We are happy you interested in our design for experiments and carefully review our paper and code. Here are the detailed response for your questions and concerns.
>
> ***
>
> **Q1**: The limitations should be discussed in the main paper, not appendix.
>
> **A1**: Thanks for the suggestion. We will move the limiation to main paper in our revised version.
>
> ***
>
> **Q2**: Baselines are all transformer-based methods. It would be interesting to compare with different architectures (e.g., try the Mamba model).
>
> **A2**: Thanks for raising this concern. We add three more baselines that are not transformer-based for comparision: TCN, ModernTCN, and Mamba. The results are presented in R-Table 1 in the attached pdf file in general response. We evaluated our method and 6 baselines across the 3 new datasets and the previously used TDBrain dataset. Our method achieved the best F1 score on 3 datasets and the second-best on the remaining one.
>
> ***
>
> **Q3**: The improvements over existing methods seems marginal. Some results are tied as they lie in the range of the standard deviation of the best model. It would be good to highlight these results as well.
>
> **A3**: Thanks for raising this concern. PatchTST outperforms our method on the PTB-XL dataset by 0.6 in F1 score. As we observe from the ablation study, PTB-XL does not benefit from cross-channel patching too much; we guess the reason is the smaller number of channels(12) of PTB-XL. In other words, we believe our model is more suitable for datasets that demonstrate more channel correlations. We will discuss and analyze more on results in our revised version of paper.
>
> ***
>
> **Q4**: The efficiency of the models should be benchmarked (e.g., training time).
>
> **A4**: We have included a table (Table R.2) in the attached PDF that presents the runtime performance of several older baselines known for their strong performance, as well as the three new baselines, on the newly added dataset: FLAAP (13,123 samples, 10 classes). In general, CNN-based methods are fast, and the Mamba is extremely slow for unknown reasons. We guess this is because Mamba's new architecture does not match the GPU hardware optimization well. Our method achieves close runing time to other transformer-based methods.
>
> ***
>
> **Q5**: Can the superior performance also extends to other vital signs and medical time-series prediction tasks? For example, PPG data and other multi-modal datasets.
>
> **A5**: Yes, we believe our model is adaptable to various types of medical time series, particularly those with multi-channel correlations. For multi-modal learning of medical time series collected simultaneously (e.g., PPG, EEG, fNIRS, ECG), we see three potential research directions for applying our method: 1) Interpolating and Concatenating: Interpolating all modalities to the same length and vertically concatenating them into a multi-channel time series. 2) Individual Representation Learning: Using separate models to learn individual representations for each modality, then concatenating these representations for downstream tasks. 3) Granularity-based Modality Correspondence: Applying different granularities to correspond to different modalities during the learning process. We plan to explore these directions in future work.
>
> Additionally, we train our method on a new dataset, MIMIC-PERform-AF, which uses PPG and ECG data to distinguish atrial fibrillation patients (20,400 samples, 2 classes). The results, presented in R-Table 1, show that our method outperforms the second-best result by approximately 3% in F1 score.

---

> ### Comment · Reviewer_zGJA · 2024-08-09
> **Main paper revision**
>
> Dear Authors,
>
> Thank you for addressing my concerns with additional material. However, a revised version of the paper must be provided to fully address the points in my and the other authors' reviews. Thank you.

---

> > ### Author Response · Authors · 2024-08-09
> >
> > Dear Reviewer,
> >
> > Thank you for your feedback on our additional material. While we would be happy to provide a revised version of the entire paper, according to the rebuttal rules of Neurips 2024, we are not allowed to upload the revision of our paper until the camera-ready stage. We are only allowed to upload one page of PDF in the global response, including tables and figures.
> >
> > For your convenience, we quote some FAQs from **NeurIPS 2024 FAQ for Authors** in the **Reviewing/Discussion process**:
> >
> > * **Can we upload a revision of our paper during the rebuttal/discussion period?** No revisions are allowed until the camera-ready stage.
> >
> > * **Can we upload a revision of the supplementary materials during the rebuttal/discussion period?** No. You may revise it for the camera-ready stage.
> >
> > * **What is the length limit in the rebuttal phase?** You can submit a rebuttal of up to 6000 characters per review, and one global rebuttal of up to 6000 reviews. These are posted by clicking the "Rebuttal" and "Author Rebuttal" buttons. You can additionally add a one-page PDF with Figures and tables. You can upload this PDF after you click the "Author Rebuttal" button.
> >
> > * **Where can I see my submitted PDF for the author-rebuttal?** There is a link to the PDF at the end of the global rebuttal.
> >
> > Could you use our current rebuttal for your assessment? We are happy to answer your additional questions. Thank you again!

---

> > > ### Comment · Reviewer_zGJA · 2024-08-11
> > >
> > > Dear Authors,
> > >
> > > Thank you for your answer. This has apparently changed this year. You have addressed my points. However, I think the current score is appropriate.

---

### Official Review · Reviewer_vj5L · 2024-07-12

**Soundness:** 3
**Presentation:** 3
**Contribution:** 2
**Rating:** 5
**Confidence:** 4

**Summary:**

This paper proposes a multi-granularity patching Transformer for medical time series classification. The proposed model, Medformer leverages cross channel patching to learn inter-channel correlations. It utilizes multi-granularity embedding for learning temporal patterns. Finally, it takes 2 stage self-attention to aggregate information from multiple granularities. The experiments on 5 datasets show the effectiveness of Medformer.

**Strengths:**

1.The paper is well structured and easy to follow.

2.The problem of medical time series classification is important.

3.The paper conducts several experiments on medical time series datasets and demonstrates the effectiveness of Medformer.

**Weaknesses:**

1.The novelty of this paper is not enough. The idea of leveraging multi-granularity and multi-channel features has been studied in time series field.

2.Time series classification has also been well studied. The paper should clarify and focus the specific challenges for medical time series classification compared to general time series.

3.For general time series classification, several new SOTA methods have been released, such as Time-LLM, ModernTCN, and Lag-llama, etc. Such recent studies should be compared in the experiments.

4.The results of Medformer are not superior in several metrics.

**Questions:**

1.How about the performance of the medformer when it is applied to general time series classification? This experiments could demonstrate the advantages of the model.

2.What are the new challenges of time series classification in medical datasets? How does Medformer resolve it?

**Limitations:**

Yes.

---

> ### Author Rebuttal · Authors · 2024-08-07
>
> Thank you for your suggestions on our paper! We are happy you feel our paper is well-structured and easy to follow. We respond to each of your questions and concerns. If you do not feel we have sufficiently justified a higher score, please let us know your further concerns and how to improve. Thank you again!
>
> ***
>
> **Q1**: The idea of leveraging multi-granularity and multi-channel features has been studied in time series field.
>
> **A1**: We agree that some methods utilized multi-granularity or multi-channel features in the time series domain, as we compared or described in the related work. However, we are the first to combine multi-granularity with cross-channel patching to model cross-channel information across different levels of granularity. In particular, we design an embedding method for medical time series data, where different channels represent signals from different regions of the same organs (brain, heart, etc.). Tight correlations among the channels are evident and valuable from a medical perspective[4][5][6]. Therefore, our approach is tailored to capture these correlations effectively with one-step data embedding. In contrast, as we mentioned in the paper (lines 107-110), existing patching methods in time series transformers, such as those inherited from PatchTST [1][2][3], rely on single-channel patching. These methods are designed for time series forecasting in tasks like weather and energy prediction, where different channels may represent entirely different meanings and, therefore, follow a channel-independent strategy. Thus, our method exhibits a significant deviation from the other methods and is a rational design for the specific datasets and challenges.
>
> ***
>
> **Q2**: The paper should focus on the challenges of medical time series classification. What are those challenges, and how does Medformer resolve them?
>
> **A2**:
> 1) **Strong correlation between channels and features at a wide frequency range motivates our design of multi-granularity cross-channel patching**: Medical time series data are collected from human subjects using sensors and electrodes, usually resulting in multi-channel and multi-modal data. As we discussed in our previous response, different channels in medical time series often represent activities in various regions of the same underlying organs, e.g., brain, heart, etc., from a medical perspective. Therefore, strong inter-channel correlations exist naturally [4][5][6]. Moreover, biomarkers in these signals appear at a wide range of frequencies with a difference of up to 1~2 magnitude [6]. To utilize these features, we design multi-granularity cross-channel patching that enables one-step data embedding, followed by a two-stage intra-inter granularity self-attention mechanism to exploit these features further.
>
> 2) **The correlation of samples from the same subject encourages us to explore a subject-independent training setup**. As we described in the preliminaries section(lines 124-135), for medical time series collected for disease diagnosis tasks, the ultimate goal is to predict the label of **Subject** instead of **Sample** to diagnose disease. In this case, samples from unseen subjects should not be utilized in the training to follow a real-world scenario. Many existing works do not consider subject ID information[7][8]. To better simulate real-world conditions, we categorize data splitting strategies into two setups: subject-dependent and subject-independent. For subject-dependent, samples are randomly split into training, validation, and test sets. For subject-independent, we split subjects into these three sets, ensuring that samples from the same subjects are exclusively contained within each set (See Figure 2). Tables 2 and 3 in the paper show how deceptively high results could be achieved with improper setup, leading to ineffective models in real-world applications.
> ***
>
> **Q3**: Several new time series classification SOTA methods have been released, such as Time-LLM, ModernTCN, Lag-llama etc. These studies should be compared.
>
> **A3**: Thank you for recommending these exceptional works. We will review them and incorporate them into the related work section in our upcoming revision. We have included experiments for ModernTCN in Table R.1 of the attached PDF. Despite our best efforts, we could not obtain results from the two LLM-based models, Time-LLM and Lag-Llama, as they are primarily designed for time series forecasting. Fine-tuning these pre-trained models for classification tasks would require more computational resources than currently available. However, to ensure a comprehensive evaluation of our model, we add two additional methods, TCN[11] and Mamba[13], also in Table R.1.
>
> ***
>
> **Q4**: How does the Medformer perform in general time series classification?
>
> **A4**: While our method is designed for medical time series using their characteristics, we believe it applies to general time series data, particularly multi-channel time series with intrinsic correlations among channels. To prove this, we add 3 new datasets to our experiments: FLAAP, UCI-HAR, and MIMIC-PERform-AF. The results are presented in R-Table 1, attached to the general response PDF. We evaluated our method and 6 baselines across the 3 new datasets and the previously used TDBrain dataset. Our method achieved the best F1 score on 3 datasets and the second-best on the remaining one.
>
> ***
>
> **Q5**: Medformer's results are not superior in several metrics.
>
> **A5**: PatchTST outperforms our method on the PTB-XL dataset by 0.6 in F1 score. Based on the ablation study presented in Table 4, cross-channel patching does not provide as much benefit as other datasets. We guess that this might caused by the smaller number of channels (12) in PTB-XL and weak channel correlation, which limit the advantages of our cross-channel patching. However, we believe the other mechanism, such as intra-inter granularity self-attention, can be combined with PatchTST to improve the result on PTB-XL.

---

> > ### Comment · Reviewer_vj5L · 2024-08-13
> > **Thanks for the rebuttal**
> >
> > Thanks for the rebuttal. The authors clarify the ideas and motivations of MedFormer and add more baselines. I have increased my score.

---

> > > ### Author Response · Authors · 2024-08-13
> > >
> > > Thank you for your reply and endorsement!

---

> ### Author Response · Authors · 2024-08-12
>
> Dear reviewer vj5L:
>
> We would like to kindly remind you that the author-reviewer discussion will end in 1 day. Could you take a look at our rebuttal and let us know if our rebuttal has addressed your concerns? As our paper is currently on the borderline, your input will be greatly appreciated.
>
> Best regards

---

> ### Comment · Area_Chair_4Q9B · 2024-08-12
> **Please respond to the rebuttal of NeurIPS Submission7556! The response period ends in one day!**
>
> Reviewer vj5L,
>
> Thank you for your service in reviewing for NeurIPS!
> Your work is not done, however, as part of a reviewer's responsibility is to engage with the authors during the discussion period.
>
> The authors of Submission 7556 (Medformer) have provided an extensive response to your review.
> Please read it and indicate the extend to which it addresses your concerns. Please indicate clearly which of the issues you raised are addressed and which are not, such that the authors have a final chance to reply.
>
> There is only one day left in the discussion period, so please do this ASAP!
>
>
> The AC for Submission 7556

---

### Official Review · Reviewer_AkiE · 2024-07-12

**Soundness:** 3
**Presentation:** 3
**Contribution:** 3
**Rating:** 6
**Confidence:** 4

**Summary:**

In this paper, the authors introduced a multi-granularity patching transformer tailored specifically for medical time series classification. To leverage the characteristics of medical time series data, the model incorporated three unique features : cross-channel patching to leverage inter-channel correlations, multi-granularity embedding for capturing features at different scales, and two-stage (intra- and inter-granularity) multi-granularity self-attention for learning features and correlations within and among granularities. The model was well tested with several datasets and showed mixed results against other models.

**Strengths:**

Considering the charactaristics of time series data in medical domain, this approach is quite reasonable and is showing strong results especially in EEG datasets.

**Weaknesses:**

Because this model has been evaluated on very specific time series data, EEG and ECG, it is unclear whether this approach can be extended to many other time series data outside the medical domain.

**Questions:**

As mentioned in Appendix G, what domains or data sets should be tested next to test the ability of this approach?

**Limitations:**

Currently the proposed model was tested only in EEG and ECG domains, thus the scope is too narrow for the general audience of the NeurIPS.

---

> ### Author Rebuttal · Authors · 2024-08-07
>
> Thank you for your feedback and concerns regarding our paper! We appreciate your thoughtful questions. Here are the reponses to your question and concern. If you feel our response does not fully justify a higher score, please let us know how we can further improve our work. Thank you again for your consideration.
>
> ***
>
> **Q1**: Extension to general time series outside the medical domain.
>
> **A1**: Thank you for raising this concern. Our answer contains two parts.
>
> 1) We agree our method is designed for medical time series due to the inherent correlations among channels from a medical perspective[4][5][6]. However, we believe our method has the potential to be extended to other domains as well. Specifically, our approach is particularly well-suited for multi-channel time series data where theoretical correlations exist among different channels. Our multi-granularity cross-channel patching effectively embeds these channel correlations in a single step. Potential domains for application include human vital signs for health monitoring and time series data collected from mobile sensors for human activity recognition.
>
> 2) To further evaluate the generalizability of our method, we add three new datasets, two human activities recognition datasets FLAAP(13123 samples, 10 classes) and UCI-HAR(10299 samples, 6 classes), and one multi-modal vital signs(PPG and ECG) dataset MIMIC-PERform-AF(20400 samples, 2 classes) for atrial fibrillation classification. See R-Table 1 for the results in the pdf file attached to the general response. In this table, we compare with three old baselines with good performance so far, Crossformer, Reformer, and Transformer, and three new baselines TCN, ModernTCN, and Mamba. We evaluate our method and the six baselines on three new datasets and one old dataset, TDBrain. Our method demonstrates top-1 Accuracy and F1 score in three datasets and second-best in the remaining one.

---

> > ### Comment · Reviewer_AkiE · 2024-08-11
> >
> > I would like to thank the authors for their rebuttal. The additional experiments clearly addressed my concerns and strongly demonstrated the advantage of the proposed method. Thus I updated my score. I look forward to reading the integrated final version.

---

> > > ### Author Response · Authors · 2024-08-11
> > >
> > > Thank you so much for your endorsement! We will integrate everything into the final version of paper.

---

### Author Rebuttal · Authors · 2024-08-07

We appreciate the thoughtful feedback provided by all the reviewers. We appreciate reviewer ApjC for recognizing the novelty of our work in being the first to integrate cross-channel patching and a new multi-granularity self-attention mechanism. In response to the reviewers' insightful comments, we have conducted additional experiments (Table R.1 in the attached PDF), including three new baselines, TCN[11], ModernTCN[12], and Mamba[13], and three new datasets, including two human activities recognition datasets, FLAAP(13123 samples, 10 classes)[14] and UCI-HAR(10299 samples, 6 classes)[15], and one multi-modal vital signs(PPG and ECG) dataset MIMIC-PERform-AF(20400 samples, 2 classes) for atrial fibrillation classification[16]. We will incorporate the newly added contents into the next version of our paper.

Thank you again for your suggestions and feedback! We work hard to refine our paper, and we sincerely hope our responses are informative and helpful. If you feel we have not sufficiently addressed your concerns to motivate increasing your score, we would love to hear from you further on what points of concern remain and how we to improve our work. Thank you again!

Due to space limitations, we include **all the references** here in the global response and **all the new tables and figures** in the attached PDF file . All the new tables and figures are start with **R** to distinguish from the submitted paper verion. The experiment results of the ablation study in R-tables 1 and 2 and R-Figure 1 a) and b) can also be found in the attached PDF file.

***

## **References**
[1] Yuqi Nie, Nam H Nguyen, Phanwadee Sinthong, and Jayant Kalagnanam. A time series is worth 64 words: Long-term forecasting with transformers. ICLR, 2023.

[2] Yunhao Zhang and Junchi Yan. Crossformer: Transformer utilizing cross-dimension dependency for multivariate time series forecasting. ICLR, 2023.

[3] Yitian Zhang, Liheng Ma, Soumyasundar Pal, Yingxue Zhang, and Mark Coates. Multi-resolution time-series transformer for long-term forecasting. In International Conference on Artificial Intelligence and Statistics, pages 4222–4230. PMLR, 2024

[4] Vincent Bazinet, Justine Y Hansen, and Bratislav Misic. Towards a biologically annotated brain connectome. Nature reviews neuroscience, 24(12):747–760, 2023.

[5] Imperatori, Laura Sophie, et al. "EEG functional connectivity metrics wPLI and wSMI account for distinct types of brain functional interactions." Scientific reports 9.1 (2019): 8894.

[6] Singh, A., Dandapat, S. (2015). Two-Dimensional Processing of Multichannel ECG Signals for Efficient Exploitation of Inter and Intra-Channel Correlation. In: Bora, P., Prasanna, S., Sarma, K., Saikia, N. (eds) Advances in Communication and Computing. Lecture Notes in Electrical Engineering, vol 347. Springer, New Delhi.

[7] Cosimo Ieracitano, Nadia Mammone, Alessia Bramanti, Amir Hussain, and Francesco C Morabito. A convolutional neural network approach for classification of dementia stages based on 2d-spectral representation of eeg recordings. Neurocomputing, 323:96–107, 2019

[8] Fangzhou Li, Shoya Matsumori, Naohiro Egawa, Shusuke Yoshimoto,Kotaro Yamashiro, Haruo Mizutani, Noriko Uchida, Atsuko Kokuryu, Akira Kuzuya, Ryosuke Kojima, et al. Predictive diagnostic approach to dementia and dementia subtypes using wireless and mobile electroencephalography: A pilot study. Bioelectricity, 4(1):3–11, 2022

[9] Rahimpour, S., Rajkumar, S. and Hallett, M. (2022) “The Supplementary Motor Complex in Parkinson’s Disease,” Journal of Movement Disorders.

[10] Helson, P. et al. (2023) “Cortex-wide topography of 1/f-exponent in Parkinson’s disease,” npj Parkinson’s Disease.

[11] Shaojie Bai, J Zico Kolter, and Vladlen Koltun. An empirical evaluation of generic convolutional and recurrent networks for sequence modeling. arXiv preprint arXiv:1803.01271, 2018.

[12] Luo, Donghao, and Xue Wang. "Moderntcn: A modern pure convolution structure for general time series analysis." ICLR. 2024.

[13] Gu, Albert, and Tri Dao. "Mamba: Linear-time sequence modeling with selective state spaces." arXiv preprint arXiv:2312.00752 (2023).

[14] KUMAR, PRABHAT; SURESH, S. (2022), “FLAAP: An open Human Activity Recognition (HAR) dataset for learning and finding the associated activity patterns ”, Mendeley Data, V1, doi: 10.17632/bdng756rgw.1

[15] Reyes-Ortiz,Jorge, Anguita,Davide, Ghio,Alessandro, Oneto,Luca, and Parra,Xavier. (2012). Human Activity Recognition Using Smartphones. UCI Machine Learning Repository. https://doi.org/10.24432/C54S4K.

[16] P. H. Charlton et al. (2022), “Detecting beats in the photoplethysmogram: benchmarking open-source algorithms,” Physiological Measurement.

---

### Decision · Program_Chairs · 2024-09-25

**Decision:**

Accept (poster)

**Comment:**

The paper presents a new transformer model for medical time series. The reviewers are unanimously positive about the paper, appreciating the combination of ideas that went into the models. Questions were raised about comparisons to state of the art models, which the authors successfully addressed during the rebuttal phase. The authors also added several datasets to demonstrate the generalizability of the method, and also explained how their method could be used to derive explanations. Overall, this is a strong paper and ready to be presented.